# PIXEL TO GAUSSIAN: ULTRA-FAST CONTINUOUS SUPER-RESOLUTION WITH 2D GAUSSIAN MODELING

**Long Peng**[1,2†]   **Anran Wu**[1†]   **Wenbo Li**[6*]   **Peizhe Xia**[1]   **Xueyuan Dai**[3]   **Xinjie Zhang**[4]
Xin Di[1]   Haoze Sun[5]   Renjing Pei[2]   Yang Wang[1,3]   Yang Cao[1*]   Zheng-Jun Zha[1*]
[1]USTC   [2]Huawei Noah's Ark Lab   [3]Chang'an University   [4]HKUST   [5]THU   [6]CUHK
https://github.com/peylnog/ContinuousSR

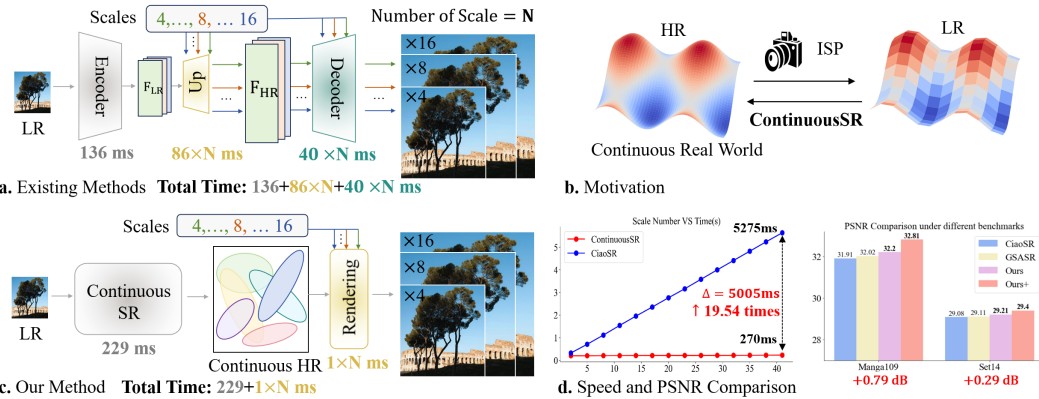

Figure 1: (a) Leveraging implicit modeling, existing ASSR methods rely on multiple upsampling and decoding steps to reconstruct HR images at different scales, which leads to low efficiency and performance. (b-d) Our method explicitly reconstructs 2D continuous HR signals from LR images in a single pass. Then, fast rendering replaces the time-consuming upsampling and decoding process to reconstruct HR images at different scales, achieving significant improvements over the state-of-the-art method GSASR in both performance (0.18 dB in Manga109) and efficiency (19.5× speedup).

## ABSTRACT

Arbitrary-scale super-resolution (ASSR) aims to reconstruct high-resolution (HR) images from low-resolution (LR) inputs with arbitrary upsampling factors using a single model, addressing the limitations of traditional SR methods constrained to fixed-scale factors (*e.g.*, × 2). Recent advances leveraging implicit neural representation (INR) have achieved great progress by modeling coordinate-to-pixel mappings. However, the efficiency of these methods may suffer from repeated upsampling and decoding, while their reconstruction fidelity and quality are constrained by the intrinsic representational limitations of coordinate-based functions. To address these challenges, we propose a novel ContinuousSR framework with a Pixel-to-Gaussian paradigm, which explicitly reconstructs 2D continuous HR signals from LR images using Gaussian Splatting. This approach eliminates the need for time-consuming upsampling and decoding, enabling extremely fast ASSR. Once the Gaussian field is built in a single pass, ContinuousSR can perform arbitrary-scale rendering in just 1ms per scale. Our method introduces several key innovations. Through statistical analysis, we uncover the Deep Gaussian Prior (DGP) and propose DGP-Driven Covariance Weighting, which dynamically optimizes covariance via adaptive weighting. Additionally, we present Adaptive Position Drifting, which refines the positional distribution of the Gaussian space based on image content, further enhancing reconstruction quality. Extensive experiments on seven benchmarks demonstrate that our ContinuousSR delivers significant improvements in SR quality across all scales, with an impressive 19.5× speedup when continuously upsampling an image across forty scales.

---

*Corresponding Authors.

†These authors contributed equally to this work.

# 1 INTRODUCTION

Cameras and smartphones discretize continuous real-world scenes into discrete 2D digital images Castleman (1979); Jain (1989); Xia et al. (2024), as illustrated in Figure 1(b). However, limitations in sensor resolution, among other factors, often lead to low-resolution (LR) images that fail to meet user requirements. Image super-resolution (SR) has been proposed to enhance image resolution and finer details Wang et al. (2020); Liu et al. (2022). Unlike traditional fixed-scale super-resolution Peng et al. (2024a); Liang et al. (2021); Chen et al. (2023c); Lim et al. (2017a); Zhang et al. (2018), which uses multiple models to learn mappings for fixed scales (*e.g.*, ×2, ×3, ×4), arbitrary-scale super-resolution (ASSR) employs a single model to handle super-resolution with arbitrary scales, which has attracted significant attention Hu et al. (2019); Wei & Zhang (2023); Chen et al. (2023a); Lee & Jin (2022a); Liu et al. (2024); Wan et al. (2024); Li et al. (2024c); Liu & Tang (2025).

Among these approaches, implicit neural representation (INR) has emerged as a leading technique, delivering visually compelling results Chen et al. (2021); Lee & Jin (2022b); Xu et al. (2021); Cao et al. (2023); Xia et al. (2024); Jiang et al. (2025). INR aims to learn a continuous mapping from pixel coordinates to pixel values, enabling arbitrary-scale super-resolution through multiple upsampling and decoding steps, as illustrated in Figure 1(a). For example, LIIF Chen et al. (2021) is the first to introduce INR into ASSR, employing multi-layer perceptrons to learn this mapping. Later, CiaoSR Cao et al. (2023) and CLIT Chen et al. (2023a) leverage Transformers to enhance the modeling of long-range dependencies in feature upsampling and decoding. However, the reconstruction fidelity and quality of these methods are inherently constrained by the representational limitations of coordinate-based implicit functions, making it challenging to effectively model continuous high-resolution signals, ultimately leading to suboptimal performance. Recent work has explored Gaussian representations for ASSR as an alternative to coordinate-based mappings. GaussianSR Hu et al. (2024) models Gaussians in the feature space but, like INR-based approaches, still requires a dedicated decoding pass for every target scale, resulting in high computational cost. GSASR Chen et al. (2025) predicts scale-conditioned 2D Gaussians in RGB space, but regenerates a separate Gaussian set for each scale, which limits both cross-scale consistency and efficiency.

Given that LR images are discretized from continuous 2D signals, we pose the fundamental question: "*Can we directly reconstruct continuous HR signals from LR images and flexibly choose the desired scale?*" As illustrated in Figure 1(b), this approach not only enhances signal continuity through continuous modeling—leading to improved reconstruction quality—but also significantly boosts efficiency by eliminating the need for time-consuming upsampling and decoding. This idea enables fast and flexible ASSR, making real-world applications more practical.

In this paper, we present the ContinuousSR framework built upon the Pixel-to-Gaussian paradigm, which reconstructs 2D continuous HR signals through Gaussian modeling. By recovering a continuous high-resolution Gaussian field, our method enables fast sampling directly from the continuous representation, effectively replacing conventional time-consuming upsampling and decoding procedures. This design leads to consistently strong reconstruction quality while requiring only 1 ms for ASSR inference, delivering a substantial improvement in efficiency.

Directly applying Gaussian modeling to simulate real-world images is highly challenging due to the intricate interweaving of pixel distributions and parameters. To address this, we first identify the Deep Gaussian Prior (DGP), revealing that the distribution of Gaussian field parameters follows a Gaussian pattern with regularities in their range, as shown in Figure 2(a-b). Leveraging this insight, we sample pre-defined Gaussian kernels from the DGP and introduce a novel DGP-Driven Covariance Weighting module, which efficiently optimizes covariance via adaptive weighting. This helps guide the model toward the global optimum. Furthermore, we propose a Adaptive Position Drifting module, which dynamically adjusts the spatial positions of Gaussian kernels based on image content, enhancing structural accuracy. With these innovations, our method not only surpasses state-of-the-art approaches by up to 0.18 dB in reconstruction performance but also achieves a 19.5× speedup when continuously upsampling across forty scales. Our main contributions are as follows:

- A novel ContinuousSR is proposed to reconstruct continuous HR signals from LR images by 2D Gaussian modeling, thereby enabling fast and high-quality super-resolution with arbitrary scale.

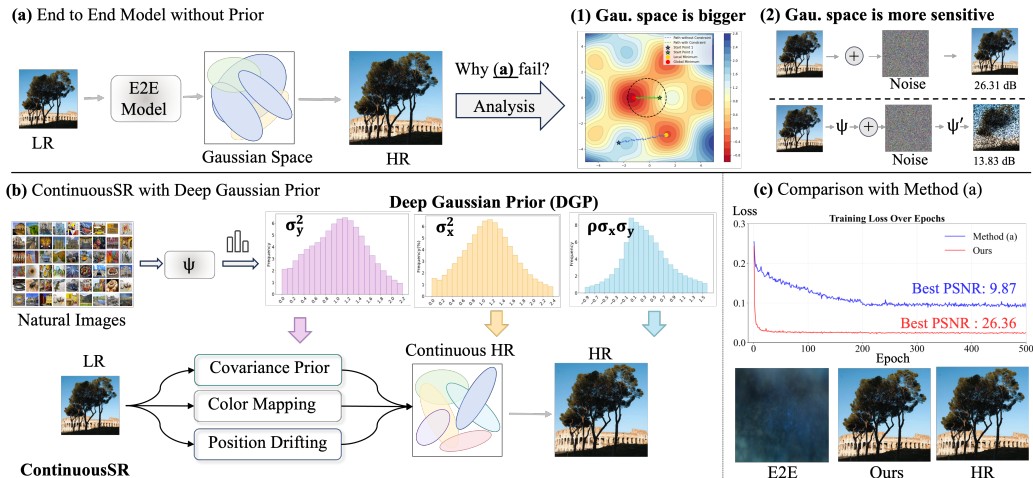

Figure 2: (a) Directly learning the end-to-end model from LR to the Gaussian field is challenging due to the vastness and sensitivity of the Gaussian space. (b-c) Through statistical analysis of 40,000 natural images, we uncover the Deep Gaussian Prior and propose Position Drifting, Covariance Prior, and Color Mapping to propose the ContinuousSR, enhancing the quality of the Gaussian field.

- The Deep Gaussian Prior (DGP) is discovered, based on which DGP-Driven Covariance Weighting is proposed to facilitate the optimization of covariance. Furthermore, Adaptive Position Drifting is introduced to dynamically learn spatial positions in Gaussian space.

- Extensive experiments demonstrate that our method achieves state-of-the-art performance on seven benchmarks and ultra-fast speed.

## 2 RELATED WORK

### 2.1 ARBITRARY-SCALE SUPER-RESOLUTION

Although traditional fixed-scale super-resolution (FSSR) methods, which use separate models to learn different super-resolution scales, have achieved significant progress Dong et al. (2014); Ledig et al. (2017); Zhang et al. (2018); Kim et al. (2016); Cavigelli et al. (2017); Zhang et al. (2021); Wang et al. (2018); Di et al. (2024); Peng et al. (2024b), they struggle to meet the demand for arbitrary-scale super-resolution in real-world scenarios. Additionally, maintaining multiple models incurs high computational costs, making them less practical. To address these limitations, Arbitrary-Scale Super-Resolution (ASSR) has been proposed to achieve it with a single model, gaining increasing attention in recent years Cao et al. (2023); Hu et al. (2019); Fu et al. (2024); He & Jin (2024b); Zhu et al. (2025); Zhao et al. (2024); Tsai et al. (2024); Fu et al. (2024); Zhang et al. (2024a); Shang et al. (2024); Jiang et al. (2024); Duan et al. (2024); He & Jin (2024a); Jiang et al. (2025); Liu & Tang (2025). For example, MetaSR Hu et al. (2019) was the first to introduce the meta-upscale module to achieve arbitrary-scale super-resolution, demonstrating promising results. Inspired by the success of implicit neural representation (INR) in 3D reconstruction, LIIF Chen et al. (2021) was the first to adapt INR to super-resolution by using a multilayer perceptron to learn the mapping from image coordinates and features to RGB values. To capture more high-frequency details, LTE Lee & Jin (2022b) encodes textures in the Fourier space, while SRNO Wei & Zhang (2023) leverages neural operators to model global relationships. CLIT Chen et al. (2023a) introduces a cross-scale interaction mechanism to enhance feature learning by integrating information across different resolutions. CiaoSR Cao et al. (2023) further improves long-range modeling capability by introducing transformers to INR, achieving state-of-the-art performance. LMF He & Jin (2024b) enhances local texture details by combining multi-frequency information in a computationally efficient manner, significantly reducing computational costs while maintaining the reconstruction of fine-grained features. However, these implicit modeling methods struggle to explicitly reconstruct continuous HR signals and require time-consuming upsampling and decoding, leading to low performance and efficiency.

## 2.2 GAUSSIAN SPLATTING

Gaussian Splatting (GS) is introduced into 3D as a faster, more efficient alternative to NeRF, using anisotropic 3D Gaussians for real-time rendering and direct scene manipulation Kerbl et al. (2023). Building on 3DGS, 2D Gaussian Splatting improves the geometric accuracy of radiance fields by combining 2D Gaussians with precise scene projections Huang et al. (2024). Recently, 2DGS finds applications in image processing Dong et al. (2025). For instance, GaussianImage Zhang et al. (2024b) proposes leveraging GS for image compression and reconstruction through long-time optimization of GS parameters, while GaussianSR Hu et al. (2024) applies Gaussian Splatting in the feature space to enhance visual quality and speed, and GSASR Chen et al. (2025) predicts scale-conditioned Gaussians for ASSR. However, these methods still struggle to reconstruct continuous HR signals and suffer from long optimization times or repeated upsampling and decoding.

## 3 MOTIVATION

To capture the real world, advanced imaging sensors (*e.g.*, CMOS) are used to project the 3D continuous world into 2D and then discretized 2D continuous signals into 2D discrete signals Castleman (1979); Jain (1989); Xia et al. (2024), as formulated:

$$I[m, n] = f_c(m\Delta x, n\Delta y). \tag{1}$$

where $f_c(x, y)$ represents the continuous intensity function in the spatial domain $(x, y)$. The $\Delta x$ and $\Delta y$ denote the sampling step along the spatial dimensions, while $m, n \in \mathbb{Z}$ are the corresponding discrete pixel grids. $I[m, n]$ represent the discrete images. After that, the Image Signal Processor is used to quantize, process, and encode it into a digital low-resolution image $\mathbf{I}_{\text{LR}}$.

Although many methods leveraging implicit modeling have been proposed Chen et al. (2021); Lee & Jin (2022a) to achieve ASSR by constructing coordinate-to-pixel mappings, two major challenges remain. On the one hand, the aim of ASSR is to reconstruct $f_c(x, y)$, but implicit modeling makes it difficult to explicitly model high-quality continuous functions, resulting in limited performance. On the other hand, the pipeline of INR-based ASSR methods suffers from low efficiency, as follows:

$$\mathcal{F}_{LR} = \mathbb{E}(\mathbf{I}_{\text{LR}}), \quad \mathcal{F}_{HR}^s = \mathbb{U}(\mathcal{F}_{LR}, s), \quad \mathbf{I}_{\text{HR}}^s = \mathbb{D}(F_{HR}^s). \tag{2}$$

where $\mathbb{E}$, $\mathbb{U}$ and $\mathbb{D}$ represent the Encoder, Upsampling, Decoder, respectively, and $\mathcal{F}_{HR}^s$ denotes the high-resolution feature map at scale $s$. It can be observed that for different scales $s$, this method requires multiple time-consuming upsampling $\mathbb{U}$ and decoding $\mathbb{D}$ processes to reconstruct HR images $F_{HR}^s$, as shown in Figure 1(b), resulting in inefficiency. Therefore, we propose the fundamental question: "*Can we directly reconstruct continuous HR signals from LR images?*" This serves as the inverse function of imaging process Eq. 1, as illustrated in Figure 1(c). This approach would not only perform simple sampling to replace multiple upsampling and decoding but also enhance continuity, improving efficiency and performance.

## 4 PROPOSED METHOD

### 4.1 CONTINUOUS BASIS FUNCTION

Considering that the target function is continuous, it is crucial to select an appropriate continuous basis function. In this work, we choose the Gaussian function for two main reasons: a) Leveraging the Gaussian Mixture Model (GMM) Reynolds et al. (2009), any complex continuous function can be represented as a combination of several Gaussian functions, ensuring broad applicability and theoretical soundness. b) With the recent advancements in the Gaussian splatting community Fei et al. (2024); Chen & Wang (2024), the engineering efficiency and compatibility of Gaussian functions have significantly improved, making them highly suitable for practical implementation. Therefore, we use Gaussian functions $G_i(x, y)$ as fundamental continuous functions to reconstruct real 2D continuous signals $f_c(x, y)$, as shown in the following equation:

$$f_c(x, y) = \sum_{i=1}^{N} G_i(x, y), \tag{3}$$

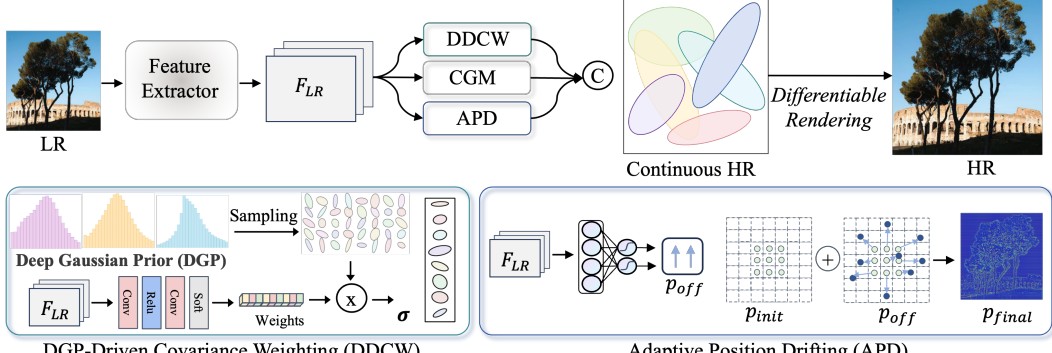

Figure 3: An overview of the proposed ContinuousSR framework, which consists of three key innovations: DGP-Driven Covariance Weighting (DDCW), Adaptive Position Drifting (APD), and Color Gaussian Mapping (CGM).

where $N$ denotes the number of Gaussian kernels, $x$ and $y$ represents the location in the 2D space. Each Gaussian kernel has eight parameters needed to optimized, which include:

$$\Sigma = \begin{bmatrix} \sigma_x^2 & \rho\sigma_x\sigma_y \\ \rho\sigma_x\sigma_y & \sigma_y^2 \end{bmatrix}, \mu = \begin{bmatrix} \mu_x \\ \mu_y \end{bmatrix}, c_{rgb} = \begin{bmatrix} c_r \\ c_g \\ c_b \end{bmatrix}, \tag{4}$$

where $c_{rgb}$ denotes the RGB parameters of each Gaussian, $\mu$ represents the position parameters, and $\Sigma$ represents the covariance matrix, resulting in a total of eight parameters to be optimized. The value of the Gaussian kernel $G_i$ at the position $(x, y)$ can be expressed as:

$$G_i(x, y, c_{rgb}, \Sigma) = c_{rgb} \frac{1}{2\pi|\Sigma_i|} \exp\left(-\frac{1}{2}d^\top\Sigma_i^{-1}d\right). \tag{5}$$

where the distance vector $d$ represents the deviations of $x$ and $y$ from their positions $\mu_x$ and $\mu_y$.

## 4.2 DIRECT END-TO-END AND DEEP GAUSSIAN PRIOR

A straightforward approach is to learn the parameters of Gaussian kernels directly from LR images through an end-to-end model. However, this approach is extremely difficult to optimize, as shown in Figure 2(a). As shown in Figure 2(c), the blue loss curve indicates that the optimization process falls into a local optimum, with the PSNR remaining as low as 10 dB. To rule out the possibility of coincidence, we conduct multiple experiments and consistently observe the same conclusion.

**Why does direct end-to-end fail?** We attribute this to two main challenges: **a) High Complexity**: Each kernel in the Gaussian space contains numerous difficult-to-learn parameters that need to be optimized, such as position, covariance, and RGB values. Their domains are unbounded, making the solution space extremely large. For instance, the covariance matrix only needs to be positive definite theoretically. This makes the Gaussian Space significantly larger than traditional image space, while introducing more local traps. This leads to a higher risk of local optima, as illustrated in Figure 2 (a). **b) High Sensitivity**: Small parameter changes in Gaussian Space can significantly impact the entire image, unlike image space where a pixel only affects itself. To further verify this, we add noise with the same distribution to both image space and Gaussian Space to evaluate sensitivity. Note that the Gaussian Space is derived through the optimization method Zhang et al. (2024b), denoted $\psi$, which requires approximately 1 minute of GPU time per scene. As shown in Figure 2(c), the PSNR drops from 26.31 dB in image space to 13.83 dB in Gaussian Space. This demonstrates that Gaussian Space is much more sensitive, making the optimization more challenging.

**Observation and Deep Gaussian Prior.** To uncover the secrets of the Gaussian Space, we conduct statistical experiments to analyze the distribution of Gaussian parameters. Specifically, we collect and crop approximately 40,000 high-resolution images Timofte et al. (2017); Lim et al. (2017b), and transform them into the Gaussian space using $\psi$, with optimized over 700 GPU-hours. Subsequently, we statistically analyze the key parameters of Gaussian kernels, including $\sigma_x^2$, $\sigma_y^2$, and $\rho\sigma_x\sigma_y$. The results, as shown in Figure 2(b), indicate that the distribution of most covariances is traceable: a) Approximately 99% of $\sigma_x^2$, $\sigma_y^2$, and $\rho\sigma_x\sigma_y$ fall within the ranges of $0 \sim 2.4$, $0 \sim 2.2$, and $-0.9 \sim 1.5$, respectively. b) The distributions of the three covariances generally follow a

Gaussian distribution. We define this finding as the **Deep Gaussian Prior (DGP)**, which provides valuable information to reduce the difficulty of optimization. Based on these observations, we propose an innovative method, ContinuousSR, which for the first time achieves representation learning from low-resolution (LR) images to continuous HR signals. Specifically, ContinuousSR introduces DGP-Driven Covariance Weighting, which simplifies the optimization difficulty in Gaussian Space by constructing pre-defined Gaussian kernels, employing an adaptive weighting mechanism, and incorporating Adaptive Position Drifting based on offset drifting. This approach enables superior performance and achieves fast super-resolution results, as shown in Figure 1 and 3.

**Disccusion.** It is important to note that the Deep Gaussian Prior (DGP) is a fixed statistical prior extracted once from large-scale natural-image distributions. Throughout training, this prior remains unchanged—it is neither fine-tuned nor updated. By serving as a stable reference, DGP regularizes the covariance-learning process and steers the model toward well-conditioned and meaningful solutions. In our experiments, removing the predefined DGP dictionary and directly learning covariance parameters resulted in training instability and noticeable degradation in reconstruction quality, underscoring the critical role of DGP in ensuring convergence and robustness. Moreover, to further assess the generalization capability of DGP across broader scenarios, we conducted additional experiments and analyses in Section 5.2.

### 4.3 DGP-Driven Covariance Weighting

Directly learning Gaussian covariance parameters remains challenging due to their unknown range and sensitive space. Therefore, we propose a novel DGP-Driven Covariance Weighting, which leverages the deep Gaussian prior (DGP) to construct a set of pre-defined Gaussian kernels. This approach simplifies the task of directly learning covariance parameters into learning a set of weighting coefficients to combine the pre-defined kernels and represent the target kernel, as shown in Figure 3.

Specifically, using the DGP, we sample the three covariance parameters $\sigma_x^2$, $\sigma_y^2$, and $\rho$ from the corresponding distributions of the DGP. These parameters are then used to construct a dictionary of $N$ pre-defined Gaussian kernels. The sampling process is expressed as:

$$\sigma_{i,x}^2, \sigma_{i,y}^2 \sim \mathcal{P}(\sigma_x^2), \mathcal{P}(\sigma_y^2); \quad \rho_i \sigma_{i,x} \sigma_{i,y} \sim \mathcal{P}(\rho \sigma_x \sigma_y), \tag{6}$$

to construct pre-defined Gaussian kernels $\mathcal{K}$:

$$\mathcal{K} = \{G_i(\begin{bmatrix} \sigma_{i,x}^2 & \rho_i \sigma_{i,x} \sigma_{i,y} \\ \rho_i \sigma_{i,x} \sigma_{i,y} & \sigma_{i,y}^2 \end{bmatrix})\}_{i=1}^N. \tag{7}$$

These candidate covariance kernels cover the majority of types and ranges commonly observed in natural images, providing valuable prior information to facilitate network convergence. We then extract features $\mathcal{F}_{\text{LR}}$ from the input LR image $\mathbf{I}_{\text{LR}}$ using the backbone encoder $\mathbb{E}$. Then, to adaptively generate the target covariance kernel, we introduce an adaptive weighting mechanism that learns a set of weights $\mathbf{W} = \{w_i\}_{i=1}^N$ based on the extracted features. These weights are computed by the adaptive weighting module $\mathcal{M}_{\text{weight}}$, which operates as follows:

$$\mathbf{W} = \text{Softmax}(\mathcal{M}_{\text{weight}}(\mathcal{F}_{\text{LR}})). \tag{8}$$

The adaptive weighting module $\mathcal{M}_{\text{weight}}$ is implemented using several layers of convolutional neural networks (CNNs). Finally, each target kernel is generated by performing a weighted combination of the pre-defined kernels in the dictionary:

$$G_{\text{target}} = \sum_{i=1}^N w_i \cdot G_i. \tag{9}$$

Through the proposed method, we achieve effective optimization of Gaussian covariance, providing stronger prior knowledge and avoiding the local optima observed in method (a), as demonstrated in the ablation study in Section 4.2.

### 4.4 Adaptive Position Drifting

The position parameters are also critical for Gaussian kernels, as they determine their locations in the 2D space. Directly learning these positions is highly challenging, as demonstrated in Section 4.2.

Table 1: PSNR performance comparison with state-of-the-art methods under different benchmarks. Average Time (AT) is reported in milliseconds (ms). The best and the second-best results are in **bold** and bold. More comparisons are in Appendix Section A.4.

| PSNR↑ | Methods | ×4 | ×6 | ×8 | ×10 | ×12 | ×16 | ×18 | ×20 | ×32 | ×48 | AT |
|---|---|---|---|---|---|---|---|---|---|---|---|---|
| Urban100 | MetaSR | 26.76 | 24.31 | 22.92 | 22.02 | 21.31 | 20.35 | 19.96 | 19.65 | 18.38 | 17.48 | 41.4 |
| | LIIF | 26.68 | 24.20 | 22.79 | 21.84 | 21.15 | 20.19 | 19.80 | 19.51 | 18.30 | 17.45 | 110.0 |
| | LTE | 27.24 | 24.62 | 23.17 | 22.23 | 21.50 | 20.47 | 20.06 | 19.77 | 18.47 | 17.52 | 151.8 |
| | SRNO | 26.98 | 24.43 | 23.02 | 22.06 | 21.36 | 20.35 | 19.95 | 19.67 | 18.39 | 17.51 | 65.7 |
| | CiaoSR | 27.42 | 24.84 | 23.34 | 22.34 | 21.60 | 20.54 | 20.11 | 19.77 | 18.45 | 17.51 | 341.5 |
| | MambaSR | 27.02 | 24.44 | 23.01 | 22.06 | 21.36 | 20.34 | 19.95 | 19.65 | 18.29 | 17.48 | 90.5 |
| | GaussianSR | 26.20 | 23.76 | 22.35 | 21.38 | 20.66 | 19.68 | 19.31 | 19.03 | 17.86 | 17.07 | 321.4 |
| | GSASR | 27.56 | 24.94 | 23.44 | 22.42 | 21.68 | 20.6 | 20.14 | 19.81 | 18.52 | 17.45 | 89.1 |
| | Ours | 27.65 | 25.02 | 23.53 | 22.54 | 21.79 | 20.72 | 20.28 | 19.93 | 18.61 | 17.60 | **3.3** |
| | Ours+ | **28.22** | **25.43** | **23.87** | **22.86** | **22.08** | **20.95** | **20.54** | **20.21** | **18.77** | **17.70** | 4.6 |
| DIV2K | MetaSR | 29.33 | 27.03 | 25.66 | 24.69 | 23.94 | 22.82 | 22.39 | 22.01 | 20.42 | 19.25 | 123.5 |
| | LIIF | 29.27 | 26.99 | 25.60 | 24.63 | 23.89 | 22.77 | 22.34 | 21.94 | 20.36 | 19.19 | 480.6 |
| | LTE | 29.50 | 27.20 | 25.81 | 24.84 | 24.09 | 22.94 | 22.50 | 22.12 | 20.50 | 19.31 | 1407.5 |
| | SRNO | 29.42 | 27.12 | 25.74 | 24.77 | 24.03 | 22.90 | 22.46 | 22.06 | 20.47 | 19.27 | 390.9 |
| | CiaoSR | 29.59 | 27.28 | 25.89 | 24.91 | 24.15 | 22.99 | 22.54 | 22.16 | 20.50 | 19.30 | 1857.8 |
| | MambaSR | 29.36 | 27.08 | 25.70 | 24.74 | 23.99 | 22.87 | 22.44 | 22.05 | 20.46 | 19.27 | 398.3 |
| | GaussianSR | 29.03 | 26.73 | 25.29 | 24.23 | 23.44 | 22.26 | 21.81 | 21.42 | 19.90 | 18.76 | 4962.8 |
| | GSASR | 29.63 | 27.30 | 25.91 | 24.91 | 24.14 | 22.97 | 22.52 | 22.13 | 20.47 | 19.25 | 434.1 |
| | Ours | 29.71 | 27.40 | 26.00 | 25.02 | 24.25 | 23.07 | 22.65 | 22.25 | 20.56 | 19.37 | **3.5** |
| | Ours+ | **29.80** | **27.47** | **26.07** | **25.08** | **24.33** | **23.18** | **22.74** | **22.35** | **20.68** | **19.45** | 4.7 |
| LSDIR | MetaSR | 26.54 | 24.64 | 23.54 | 22.79 | 22.24 | 21.42 | 21.09 | 20.80 | 19.62 | 18.68 | 50.4 |
| | LIIF | 26.49 | 24.59 | 23.49 | 22.75 | 22.21 | 21.40 | 21.09 | 20.75 | 19.59 | 18.65 | 226.4 |
| | LTE | 26.73 | 24.78 | 23.65 | 22.88 | 22.33 | 21.48 | 21.15 | 20.85 | 19.66 | 18.71 | 451.5 |
| | SRNO | 26.65 | 24.72 | 23.61 | 22.85 | 22.30 | 21.45 | 21.12 | 20.83 | 19.64 | 18.69 | 163.6 |
| | CiaoSR | 26.80 | 24.84 | 23.69 | 22.92 | 22.35 | 21.48 | 21.14 | 20.84 | 19.63 | 18.67 | 1289.3 |
| | MambaSR | 26.62 | 24.69 | 23.59 | 22.83 | 22.28 | 21.44 | 21.11 | 20.82 | 19.64 | 18.70 | 197.7 |
| | GaussianSR | 26.25 | 24.39 | 23.28 | 22.49 | 21.92 | 21.06 | 20.74 | 20.45 | 19.29 | 18.38 | 1284.3 |
| | GSASR | 26.88 | 24.88 | 23.73 | 22.94 | 22.36 | 21.49 | 21.15 | 20.85 | 19.61 | 18.62 | 312.4 |
| | Ours | 26.95 | 24.98 | 23.85 | 23.06 | 22.48 | 21.59 | 21.29 | 20.98 | 19.71 | 18.75 | **3.3** |
| | Ours+ | **27.14** | **25.07** | **23.91** | **23.13** | **22.54** | **21.69** | **21.35** | **21.06** | **19.79** | **18.82** | 4.6 |

Since each LR pixel typically corresponds to multiple pixels in the HR image, a straightforward solution is to fix the positions at the centers of the LR pixels. While this strategy simplifies the optimization process, it significantly limits the model's representational capacity, making it difficult to adaptively learn the position distribution based on image content.

To address the above issues, we propose a novel method, Adaptive Position Drifting (APD), which not only ensures efficient optimization but also improves representational capacity by allowing the model to adaptively learn positions, as shown in Figure 3. Specifically, we use the center positions of LR pixels as the initialized positions $P_{init}$ and further introduce a dynamic offset mechanism, which learns a dynamic offset from LR features $\mathcal{F}_{LR}$ by $\mathcal{M}_{pos}$ model to adjust the spatial positions. Here, we set the offset range from $-1 \sim 1$ by the Tanh activate function and add the offset $P_{off}$ to the initialized LR center positions to obtain the final positions $P_{final}$, as expressed by the following equation:

$$P_{off} = \text{Tanh}(\mathcal{M}_{pos}(\mathcal{F}_{LR})), \quad P_{final} = P_{init} + P_{off}. \tag{10}$$

where $\mathcal{M}_{pos}$ is implemented using five multilayer perceptron layers. This $P_{off}$ enables the network to adaptively learn kernel positions based on image content, resulting in denser kernel placement in regions with richer textures and enhancing the network's performance, as demonstrated in Figure 3, Section 5.3 and Appendix Section A.7.

In addition, since the RGB is range from 0 to 1 and is relatively easy to optimize, we introduce a simple Color Gaussian Mapping (CGM) to learn the RGB parameters. Specifically, this mapping is implemented using 5 multilayer perceptron (MLP) layers applied to $\mathcal{F}_{LR}$. In summary, the above three components construct our proposed ContinuousSR framework, as shown in Figure 3.

## 5 EXPERIMENT AND ANALYSIS

### 5.1 EXPERIMENT SETTING

**Datasets.** We use the commonly employed DIV2K and DF2K high-quality dataset Wang et al. (2021) as HR images, which are degraded using bicubic to generate LR for training. For evalua-

Table 2: Performance comparison of the Urban100 benchmark on SSIM, FID, and DISTS metrics.

| Metrics | Methods | ×4 | ×6 | ×8 | ×10 | ×12 | ×16 | ×18 | ×20 | ×32 | ×48 |
|---|---|---|---|---|---|---|---|---|---|---|---|
| SSIM↑ | LIIF | 0.7911 | 0.6861 | 0.6148 | 0.5642 | 0.5270 | 0.4790 | 0.4617 | 0.4503 | 0.4106 | 0.3918 |
| | LTE | 0.8069 | 0.7045 | 0.6321 | 0.5810 | 0.5422 | 0.4900 | 0.4710 | 0.4588 | 0.4145 | 0.3931 |
| | GaussianSR | 0.7751 | 0.6633 | 0.5867 | 0.5334 | 0.4967 | 0.4521 | 0.4369 | 0.4277 | 0.3969 | 0.3835 |
| | CiaoSR | 0.8110 | 0.7126 | 0.6415 | 0.5887 | 0.5503 | 0.4974 | 0.4777 | 0.4637 | 0.4168 | 0.3921 |
| | GSASR | 0.8151 | 0.7153 | 0.6428 | 0.5877 | 0.5483 | 0.4933 | 0.4708 | 0.4562 | 0.4125 | 0.3919 |
| | Ours | 0.8211 | 0.7250 | 0.6540 | 0.6010 | 0.5600 | 0.5020 | 0.4820 | 0.4680 | 0.4180 | 0.3940 |
| | Ours+ | **0.8292** | **0.7343** | **0.6624** | **0.6089** | **0.5683** | **0.5097** | **0.4893** | **0.4746** | **0.4216** | **0.3958** |
| DISTS↓ | LIIF | 0.1611 | 0.2178 | 0.2589 | 0.2926 | 0.3209 | 0.3659 | 0.3835 | 0.3990 | 0.4678 | 0.5322 |
| | LTE | 0.1570 | 0.2126 | 0.2541 | 0.2872 | 0.3157 | 0.3611 | 0.3799 | 0.3960 | 0.4695 | 0.5362 |
| | GaussianSR | 0.1740 | 0.2374 | 0.2890 | 0.3296 | 0.3631 | 0.4109 | 0.4302 | 0.4466 | 0.5157 | 0.5713 |
| | CiaoSR | 0.1533 | 0.2074 | 0.2453 | 0.2771 | 0.3049 | 0.3510 | 0.3701 | 0.3863 | 0.4513 | 0.4998 |
| | GSASR | 0.1474 | 0.1996 | 0.2366 | 0.2678 | 0.2957 | 0.3411 | 0.3607 | 0.3784 | 0.4472 | **0.4916** |
| | Ours | 0.1415 | 0.1949 | 0.2333 | 0.2640 | 0.2909 | 0.3368 | 0.3556 | 0.3727 | 0.4456 | 0.5160 |
| | Ours+ | **0.1356** | **0.1901** | **0.2299** | **0.2601** | **0.2860** | **0.3324** | **0.3504** | **0.3670** | **0.4439** | 0.5144 |
| FID↓ | LIIF | 4.76 | 24.87 | 50.05 | 77.54 | 102.47 | 145.25 | 164.41 | 179.44 | 256.95 | 311.09 |
| | LTE | 3.84 | 21.01 | 45.15 | 70.56 | 92.85 | 136.54 | 156.24 | 170.92 | 253.52 | 296.27 |
| | GaussianSR | 5.64 | 29.30 | 57.00 | 89.25 | 120.02 | 166.20 | 181.59 | 202.18 | 264.27 | 315.97 |
| | CiaoSR | 3.74 | 20.48 | 43.25 | **58.60** | 92.58 | 133.77 | 151.70 | 168.89 | 247.84 | 294.49 |
| | GSASR | 4.17 | 22.40 | 42.19 | 66.05 | 87.79 | 126.72 | 142.84 | 161.59 | 234.27 | **274.15** |
| | Ours | 3.06 | 17.33 | 39.00 | 61.73 | 82.66 | 122.12 | 136.84 | 150.70 | 227.02 | 295.34 |
| | Ours+ | **2.91** | **16.50** | **37.09** | 58.83 | **78.72** | **116.30** | **130.32** | **143.52** | **216.21** | 281.27 |

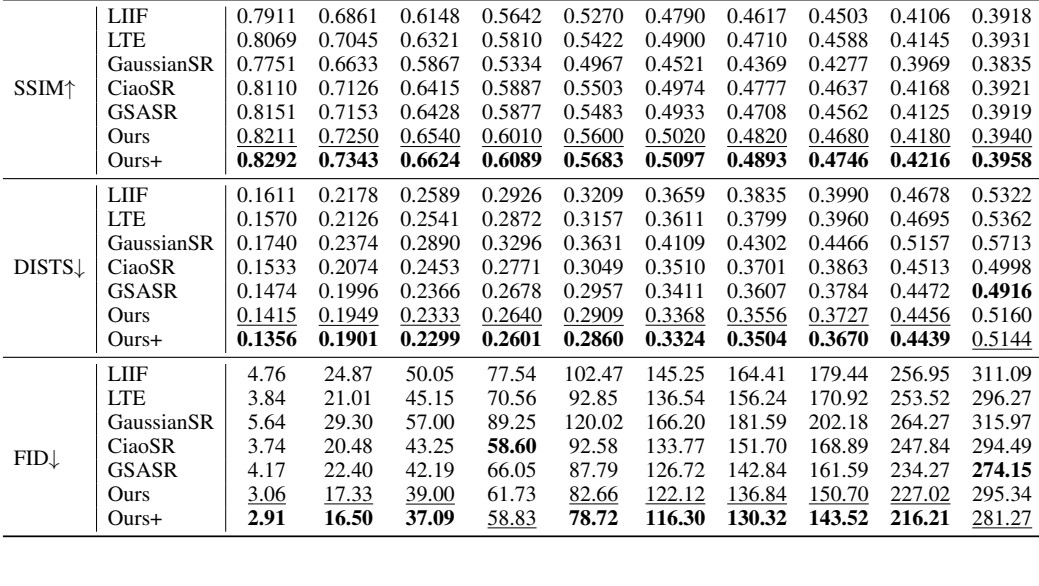

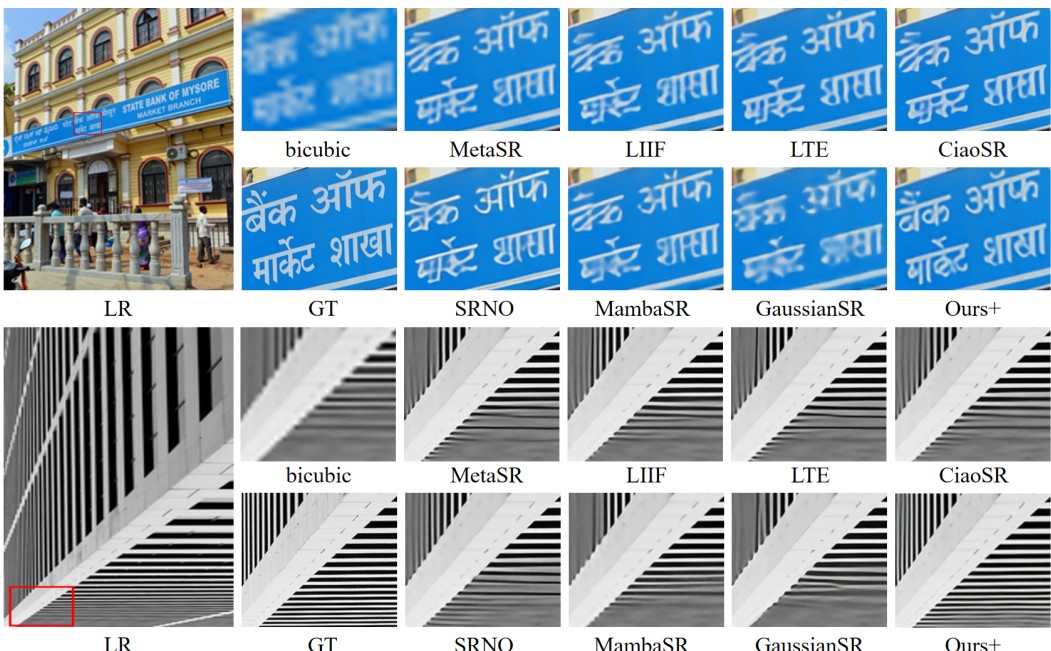

Figure 4: Qualitative comparison. The visual quality of our method outperforms existing methods. Please zoom in for a better view.

tion, we adopt Set5 Bevilacqua et al. (2012), Set14 Zeyde et al. (2012), B100 Martin et al. (2001), Urban100 Huang et al. (2015), Manga109 Matsui et al. (2017), DIV2K validation Agustsson & Timofte (2017) and LSDIR Li et al. (2023b).

**Evaluation metrics.** Following previous work Chen et al. (2021), we use PSNR, SSIM Wang et al. (2004), FID Heusel et al. (2017), and DISTS Ding et al. (2020) for evaluation. Note that the PSNR/SSIM value is calculated on the RGB channels for the DIV2K validation set and on the Y channel (i.e., luminance) of the transformed YCbCr space for the other benchmark test sets.

**Implementation details.** We conduct experiments under two training settings. First, to ensure a fair comparison with prior ASSR methods Chen et al. (2021); Lee & Jin (2022b), we adopt a configuration closely aligned with previous works. Specifically, we use SwinIR Liang et al. (2021) as the

backbone and train on the DIV2K dataset. Ground-truth (GT) images are cropped into $256 \times 256$ patches, and the corresponding LR images are generated by bicubic downsampling with scale factors sampled from a uniform distribution $U(4, 8)$. The model is optimized with Adam Kingma (2014) starting from a learning rate of $1 \times 10^{-4}$, decayed by a factor of 0.5 every 100 epochs. We train for 1000 epochs using a total batch size of 128 on 8 H20 GPUs with the L1 loss Chen et al. (2021). Second, to further explore the potential of our approach with a stronger backbone, we conduct experiments with HAT Chen et al. (2023c). In this setting, we replace the training dataset with DF2K and employ both L1 loss and frequency loss Cui et al. (2023) for supervision, and use the total batch size of 64, while keeping all other training configurations identical. We denote this stronger configuration as **Ours+** in all tables and figures.

**Compared methods.** We compare with nine state-of-the-art and popular models: MetaSR Hu et al. (2019), LIIF Chen et al. (2021), LTE Lee & Jin (2022b), ITSRN Yang et al. (2021), SRNO Wei & Zhang (2023), CiaoSR Cao et al. (2023), MambaSR Yan et al. (2024), GaussianSR Hu et al. (2024) and GSASR Chen et al. (2025). Details are provided in Appendix Section A.1.

## 5.2 QUANTITATIVE AND QUALITATIVE RESULTS

**Quantitative comparisons.** As shown in Table 1 and Table 2, our method achieves the best performance compared to existing approaches across all evaluation metrics and benchmarks. For example, on the Urban100 dataset, our method surpasses the current state-of-the-art (GSASR) by 0.09 dB in the $\times 4$, representing a substantial improvement. Similarly, in terms of SSIM and FID, our method achieves further gains in the $\times 4$ scenario, surpassing the current state-of-the-art by 0.0060 and 0.68, respectively. These results demonstrate the effectiveness and superiority of the proposed method.

**Complexity comparisons.** We present comparisons of runtime and memory usage. Specifically, we evaluate the average runtime across 45 different scales, ranging from $\times 4$ to $\times 48$. As shown in Table 1, our method significantly outperforms existing methods in terms of speed. As described above, the key advantage of our framework is that the continuous high-resolution (HR) signal is generated only once, after which super-resolution at all target scales is achieved through fast rendering. Since methods such as MetaSR must regenerate scale-specific features for every target scale, their average runtime is significantly slower. In contrast, our method performs only the lightweight rendering step (approximately 1 ms) for each scale, which leads to the extremely fast average runtime reported. For instance, it surpasses the current state-of-the-art method (GSASR) by nearly 124 times on the DIV2K dataset. Moreover, we also provide comparisons of memory usage. Specifically, we set the input size to $48 \times 48$, disable the tiling strategy, and test the memory usage under different scales. As shown in Table 4, thanks to our efficient pipeline design, our method maintains minimal computational overhead across different scales. In contrast, existing INR-based ASSR methods, such as LIIF and CiaoSR, fail to handle larger scales and encounter OOM (out of memory) on V100 GPUs.

**Qualitative comparisons.** We present qualitative comparisons, as shown in Figure 4. Compared to existing methods, our approach reconstructs more visually pleasing details that are consistent with GT. For instance, in the bottom part of Figure 4, our method effectively reconstructs the texture details inside the building. This highlights the superiority of our method in generating realistic and perceptually satisfying results. More visual comparisons, user studies, benchmark results, FLOPs comparisons, and details are provided in Appendix Sections A.4, A.3, and A.6.

**Generalization to Out-of-Distribution Data.** To further evaluate the generalization ability of ContinuousSR beyond natural images, we conducted experiments on **medical images** using the BRATS dataset Menze et al. (2014). The low-resolution inputs were generated through bicubic degradation, and all methods were trained and tested under the same HAT backbone for fair comparison. As shown in Table 3, our method maintains strong reconstruction performance even without fine-tuning, indicating good generalization capability to out-of-distribution data.

Table 3: Generalization test of different methods on medical images (BRATS dataset).

|  | $\times 4$ | | $\times 8$ | |
| --- | --- | --- | --- | --- |
|  | PSNR | SSIM | PSNR | SSIM |
| GSASR | 28.02 | 0.8691 | 23.38 | 0.6912 |
| **Ours** | **29.93** | **0.8865** | **25.61** | **0.7373** |

Table 4: Memory usage (G) comparison.

| Memory Usage | ×4 | ×6 | ×8 | ×12 | ×16 |
|---|---|---|---|---|---|
| LIIF | 4.12 | 6.49 | 9.79 | 19.27 | **OOM** |
| CiaoSR | 12.17 | 22.83 | **OOM** | **OOM** | **OOM** |
| Ours | **2.48** | **2.49** | **2.50** | **2.52** | **2.54** |

Table 5: Ablation studies.

| DDCW | APD | PSNR | $P_{\text{init}}$ | $P_{\text{off}}$ | PSNR | $K$ | PSNR |
|---|---|---|---|---|---|---|---|
| ✓ |  | 10.5 | ✓ |  | 27.8 | $\mathcal{K}_1$ | 27.7 |
|  | ✓ | 12.3 |  | ✓ | 10.5 | $\mathcal{K}_2$ | 27.1 |
| ✓ | ✓ | **28.2** | ✓ | ✓ | **28.2** | $\mathcal{K}_{DCP}$ | **28.2** |

## 5.3 Ablation Study

In this section, we present ablation studies on our proposed APD, DDCW, and DGP using the Urban100 ×4 dataset. Specifically, we independently remove APD and DDCW, as illustrated in the right part of Table 5. The exclusion of these modules significantly exacerbates the optimization difficulty, resulting in a considerable decline in PSNR. Then, we validate the effectiveness of $P_{\text{init}}$ and $P_{\text{off}}$ in APD. As shown in the middle of Table 5, the results demonstrate that the model achieves the best representational capacity and performance when both components are employed. Finally, we evaluate the effectiveness of DGP in DDCW $\mathcal{K}_{DCP}$. Specifically, we remove the DGP and separately modify the covariance range to [0,1] and [0,10], using uniform sampling to construct $\mathcal{K}_1$ and $\mathcal{K}_2$. As shown in the right in Table 5, incorporating DGP provides a better basis function, thereby enhancing performance. More ablation studies are provided in Appendix Sections A.5.

## 6 Future Work

It is well known that, in real-world scenarios, image degradation is not limited to low resolution but often includes other types of degradation, such as rain and noise. The goal of low-level vision is to remove these degradations while enhancing image resolution and quality. To this end, we evaluate our method on Rain200H Yang et al. (2017), simulating low-resolution rainy images with bicubic downsampling. We compare our ap-

Table 6: Performance comparison under more challenging low-resolution and rainy conditions.

| Methods | ×4 | ×5 | ×6 | ×7 | ×8 |
|---|---|---|---|---|---|
| LIIF | 24.04 | 23.53 | 23.09 | 22.70 | 22.39 |
| GaussianSR | 24.04 | 23.51 | 23.08 | 22.69 | 22.38 |
| CiaoSR | 23.84 | 23.45 | 23.01 | 22.66 | 22.33 |
| Ours | **24.51** | **23.95** | **23.48** | **23.07** | **22.76** |

proach with three existing state-of-the-art methods to validate its effectiveness. As shown in Table 6, our method removes rain degradations more effectively while enhancing resolution and details, outperforming existing methods. This demonstrates the potential of our approach for other low-level vision tasks. In future work, we aim to extend it to more tasks to further enhance its applicability.

Considering that the experimental pipeline for DGP is relatively labor-intensive, we will further streamline the overall process, including exploring whether Gaussian regularities can be directly derived from more physical imaging data to simplify the pipeline. In addition, we will also work on further enhancing the network efficiency in future work, making the framework easier to deploy across a wider range of real-world applications.

## 7 Conclusion

We introduce ContinuousSR, a novel Pixel-to-Gaussian methods designed for fast and high-quality arbitrary-scale super-resolution. By explicitly reconstructing 2D continuous HR signals from LR images using Gaussian Splatting, ContinuousSR significantly improves both efficiency and performance. Through statistical analysis, we uncover the Deep Gaussian Prior (DGP) and propose a DGP-driven Covariance Weighting mechanism along with an Adaptive Position Drifting strategy. These innovations improve the quality and fidelity of the reconstructed Gaussian fields. Experiments on seven popular benchmarks demonstrate that our method outperforms state-of-the-art methods in both quality and speed, achieving a 19.5× speed improvement and 0.18dB PSNR improvement, making it a promising solution for ASSR tasks.

## Acknowledgments

This work was supported by the Natural Science Foundation of China under Grants 62225207, 62436008 and 62206262.

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

# A  APPENDIX

## A.1  DETAILS OF COMPARED METHODS

To validate the effectiveness of our proposed model, we compare it against seven state-of-the-art (SOTA) and widely adopted models: MetaSR Hu et al. (2019), LIIF Chen et al. (2021), LTE Lee & Jin (2022b), SRNO Wei & Zhang (2023), CiaoSR Cao et al. (2023), MambaSR Yan et al. (2024), GaussianSR Hu et al. (2024) and GSASR Chen et al. (2025). For a fair comparison, we select the best-performing networks for each method based on their official GitHub repositories. Specifically, we use the MetaSR model based on SwinIR, the LIIF model based on RDN, the LTE model based on SwinIR, the SRNO model based on RDN, the CiaoSR model based on SwinIR, the MambaSR model based on RDN, and the GaussianSR model on EDSR-baseline.

## A.2  MORE RELATED WORK

With the rapid advancement of deep learning Yi et al. (2025; 2021a;b); Lin et al. (2025c); Gan et al. (2025b); Lu et al. (2024a); Gan et al. (2025a); Hu et al. (2025); Yang et al. (2025c;d;b); Lu et al. (2023; 2024b; 2025); Zhou et al. (2025); Li et al. (2025a); Gao et al. (2024); Lu et al. (2022); Gong et al. (2026; 2024; 2022); Ren et al. (2025); Yang et al. (2025a); Yu et al. (2025); Gao et al. (2025); Zhu et al. (2024); Lin et al. (2024b); Zheng et al. (2025b;a; 2024; 2023); Lin et al. (2024a); Zheng et al. (2022); Lin et al. (2025b;a); Xu et al. (2025a); Jin et al. (2025); Lan et al. (2025); Wu et al. (2025); Zeng et al. (2025); Xu et al. (2025b); Sun et al. (2025); Xu et al. (2024); Zhang et al. (2025a;b; 2026); He et al. (2025b;d;h;c; 2026; 2025f;e;g;i; 2024a; 2025a); Xiao et al. (2024); He et al. (2024b; 2023a;c;b); Duan et al. (2025a); Xue et al. (2025); Duan et al. (2025b); Li et al. (2023a; 2024a;b; 2025b;d;c; 2026b;a; 2025e); Gao et al. (2022); Wan et al. (2025), some recent works have also leveraged Fourier-based priors for modeling continuous high-resolution signals Liu & Tang (2025); Akita & Ukita (2025); Zhang et al. (2024a). We initially considered using Fourier basis functions for this purpose. However, the Gaussian Splatting community currently provides more mature tools and efficient rendering pipelines, which makes Gaussian functions a more practical and suitable choice for our framework. In the future, we will consider exploring the use of Fourier basis functions in future work to investigate whether they can offer improved performance.

## A.3  USER STUDY

To further assess visual quality, we conduct a user study. Ten images are randomly selected from the test datasets, ensuring diversity in image content and complexity. Fifteen participants rate the visual quality of each processed image on a scale from 0 (poor) to 10 (excellent). Each participant evaluates the images independently to ensure unbiased results. As shown in Figure 5, the results demonstrate that existing methods frequently fail to restore high-quality images, particularly in challenging regions with fine details and textures. This leads to lower user satisfaction, with average scores ranging between 4.2 and 7.0 for most competing methods. In contrast, our method achieves the highest average score of 7.7, significantly outperforming all other approaches. The superior performance of our method demonstrates its ability to produce sharper details, better texture preservation, and visually consistent results. Participants consistently note that our method outperforms others, particularly in challenging regions, further validating its effectiveness and generalization in restoring high-quality images.

## A.4  ADDTIONAL COMPARISON

**More Benchmarks.** To further demonstrate the superiority of our proposed method, we conduct experiments to compare its performance against existing methods using the same SwinIR Liang et al. (2021) backbone on the Set5 Bevilacqua et al. (2012), Set14 Zeyde et al. (2012), B100 Martin et al. (2001), Urban100 Huang et al. (2015) and Manga109 Matsui et al. (2017) datasets. As shown in Table 7, our method still achieves state-of-the-art performance across all benchmarks and scales.
**More Comparisons with GSASR.** In the main paper, we compared ContinuousSR with GSASR using the performance reported in the GSASR paper under the SwinIR backbone setting (trained on DIV2K only). However, as the GSASR authors have also released an "Ultra Performance" variant, using a larger HAT-L backbone and the large-scale SA1B dataset, we further retrained

Table 7: Performance comparison with existing methods using the same SwinIR Liang et al. (2021) backbone on the Set5 Bevilacqua et al. (2012), Set14 Zeyde et al. (2012), B100 Martin et al. (2001) and Manga109 Matsui et al. (2017) datasets. Table performance is referred to in Cao et al. (2023).

| Dataset | Scale | SwinIR | MetaSR | LIIF | ITSRN | LTE | CiaoSR | Ours |
|---|---|---|---|---|---|---|---|---|
| Set5 | ×4 | 32.72 | 32.47 | 32.73 | 32.63 | 32.81 | 32.84 | **32.93** |
| | ×6 | - | 29.09 | 29.46 | 29.31 | 29.50 | 29.62 | **29.67** |
| | ×8 | - | 27.02 | 27.36 | 27.24 | 27.35 | 27.45 | **27.55** |
| | ×12 | - | 24.82 | - | 24.79 | - | 24.96 | **25.18** |
| Set14 | ×4 | 28.94 | 28.85 | 28.98 | 28.97 | 29.06 | 29.08 | **29.18** |
| | ×6 | - | 26.58 | 26.82 | 26.71 | 26.86 | 26.88 | **26.96** |
| | ×8 | - | 25.09 | 25.34 | 25.32 | 25.42 | 25.42 | **25.54** |
| | ×12 | - | 23.33 | - | 23.30 | - | 23.38 | **23.55** |
| B100 | ×4 | 27.83 | 27.75 | 27.84 | 27.85 | 27.86 | 27.90 | **27.94** |
| | ×6 | - | 25.94 | 26.07 | 26.05 | 26.09 | 26.13 | **26.18** |
| | ×8 | - | 24.87 | 25.01 | 24.96 | 25.03 | 25.07 | **25.13** |
| | ×12 | - | 23.59 | - | 23.57 | - | 23.68 | **23.76** |
| Manga109 | ×4 | 31.67 | 31.37 | 31.71 | 31.74 | 31.79 | 31.91 | **32.19** |
| | ×6 | - | 27.29 | 27.69 | 27.72 | 27.83 | 28.01 | **28.25** |
| | ×8 | - | 24.96 | 25.28 | 25.23 | 25.42 | 25.61 | **25.80** |
| | ×12 | - | 22.35 | - | 22.47 | - | 22.79 | **22.96** |

Table 8: Comparison of PSNR (dB), FLOPs (G), and running time (ms) on the Manga109 dataset.

| | ×4 | | | ×6 | | | ×8 | | |
|---|---|---|---|---|---|---|---|---|---|
| | PSNR | FLOPs | Times | PSNR | FLOPs | Times | PSNR | FLOPs | Times |
| SwinIR-LIIF | 31.71 | 365.11 | 150 | 27.69 | 289.77 | 141 | 25.28 | 271.35 | 137 |
| SwinIR-CiaoSR | 31.91 | 1949.71 | 319 | 28.01 | 1275.95 | 256 | 25.61 | 1048.08 | 235 |
| SwinIR-GSASR | 32.02 | 395.86 | 410 | 28.10 | 142.51 | 156 | 25.69 | 142.51 | 156 |
| SwinIR-ContinuousSR | **32.19** | **280.99** | **136** | **28.25** | **125.63** | **112** | **25.80** | **79.21** | **99** |

our ContinuousSR under exactly the same Ultra Performance setting for a fairer comparison. Table 11 reports the resulting PSNR values. Under these stronger settings, our method maintains competitive performance relative to GSASR's upper bound while offering advantages in efficiency and scalability.

**More Comparisons on Real Datasets.** To further demonstrate the superiority of the proposed method in real-world scenarios, we compare it with existing methods on the real dataset COZ Fu et al. (2024) on ×5. The results, as shown in Table 9, indicate that our method consistently outperforms existing approaches in real-world scenarios, validating the superior generalization ability of our method to real-world data.

**More Complexity Comparisons.** Furthermore, we also provide comparisons of FLOPs and inference time at a single scale on the Manga109 dataset. Specifically, considering that the image shapes in the original dataset may cause other methods to run out of memory, we fix the GT shape to 288 and evaluate the FLOPs and inference time at different scales. As shown in Table 8, our method not only achieves the best performance in terms of PSNR but also maintains the lowest FLOPs and inference time at a single scale, significantly outperforming the current SOTA method, GSASR. Note that GSASR pads input images to multiples of 16 before processing, which causes its FLOPs and runtime to remain identical at ×6 (LR: $48 \times 48$) and ×8 (LR: $36 \times 36$). Moreover, Table 1 in the main text further demonstrates our superiority in total runtime across multiple scales. These results fully validate the efficiency and superiority of the proposed method.

**More Performance Metrics.** In the main text, we have provided PSNR, SSIM, FID, and DISTS metrics to demonstrate the superiority of the proposed method. Here, we further present a comparison of LPIPS performance on the Urban100 ×4 dataset. As shown in Table 10, our method achieves the best performance in terms of LPIPS. This further validates the superiority of the

Table 9: Comparison of PSNR on the COZ dataset.

| COZ | MetaSR | LIIF | LTE | LINF | SRNO | LIT | CiaoSR | LMI | Ours |
|---|---|---|---|---|---|---|---|---|---|
| PSNR | 24.39 | 24.39 | 24.4 | 24.32 | 24.4 | 24.36 | 24.38 | 24.48 | **24.68** |

Table 10: LPIPS↓ comparison for Urban100 dataset across different methods.

| LPIPS↓ | MetaSR | LIIF | MambaSR | LTE | SRN0 | CiaoSR | GaussianSR | Ours |
|---|---|---|---|---|---|---|---|---|
| Urban100 | 0.1989 | 0.2080 | 0.2073 | 0.1934 | 0.1991 | 0.1872 | 0.2285 | **0.1803** |

proposed method in perceptual quality.

**More Compared Method.** In the main text, we have compared our proposed method with 9 existing methods to demonstrate its superiority. Additionally, we include a comparison with GaussianImage Zhang et al. (2024b). Specifically, we conduct experiments on the Set5 and Set14 datasets under the ×4 scenario. Since GaussianImage is an optimization-based end-to-end algorithm, we allow this method to optimize on LR inputs and adjust the Gaussian mapping scale to perform super-resolution for comparison. As shown in Table 18, this method fails to learn the mapping from LR to HR, resulting in poor performance. Furthermore, it is worth noting that GaussianImage requires nearly 1 minute of optimization per scene on a V100 GPU, which is impractical for real-world applications.

**More Compared Methods with GS.** Several recent GS-based ASSR methods have been proposed, such as GaussianSR Hu et al. (2024) and GSASR Chen et al. (2025). GaussianSR has already been thoroughly analyzed and compared in the main text. Here, we focus on analyzing and comparing GSASR. Although GSASR has made notable progress, it is still constrained by inefficiencies caused by multiple upsampling and decoding processes across different scales. Furthermore, GSASR performs GS in the feature and image space, which makes it struggle to ensure the continuity of reconstructed images across different scales, leading to low performance. In contrast, our method leverages 2D GS modeling to reconstruct continuous HR images, enabling both fast and high-quality ASSR. Although the GSASR method has not been open-sourced, we still compare our method against the performance reported in its paper. For example, on the LSDIR benchmark, our method achieves a performance of 27.14 dB at ×4, significantly surpassing GSASR's best reported performance of 26.73 dB. This demonstrates the superiority of our method in terms of performance. Moreover, in terms of speed, our method requires only 1 ms to generate high-quality HR images across different scales, whereas GSASR takes approximately 91-1573 ms. This further highlights the ultra-fast speed of our proposed method.

**More Details in Section 5.2.** In Table 1, the Average Time (AT) is calculated by performing super-resolution on LR images across 45 different scales, ranging from ×4 to ×48, and then averaging the total runtime. For each dataset, we select a representative LR shape and downsample it by a factor of 48 to construct the input size for each dataset, ensuring that existing ASSR methods do not encounter out-of-memory (OOM) issues. Specifically, the LR size is $21 \times 13$ for Urban100, $42 \times 28$ for DIV2K, and $29 \times 19$ for LSDIR. As shown in Table 1, our method consistently achieves significant speed advantages over existing methods across different datasets and LR shapes.

## A.5 ADDITIONAL ABLATION STUDY

Due to space limitations in the main text, we provide additional ablation experiments to demonstrate the effectiveness and rationality of the proposed method. Below, we present detailed descriptions of additional ablation studies and implementation details.

**Ablation Study on $\mathcal{K}$.** In the DGP-Driven Covariance Weighting, considering the difficulty for deep learning networks to directly interpret the specific meaning of covariance, we map $\mathcal{K}$ from its original three-dimensional representation (*i.e.*, ($\sigma_x^2$, $\sigma_y^2$, and $\rho$)) to a latent representation space through a convolutional neural network. This approach facilitates better convergence and achieves improved performance. Specifically, we explore the performance when the dimension of the latent space is set to 3, 256, and 512, as shown in Table 12. It can be observed that the best performance is achieved when the dimension is set to 512, showing significant improvements compared to the

Table 11: PSNR (dB) comparison between GSASR and Ours under the Ultra Performance setting.

| Dataset | Method | ×4 | ×6 | ×8 | ×10 | ×12 | ×16 | ×18 | ×20 | ×32 | ×48 |
|---------|--------|------|------|------|------|------|------|------|------|------|------|
| LSDIR | GSASR | 27.17 | 25.10 | 23.95 | 23.13 | 22.53 | 21.64 | 21.28 | 20.98 | 19.67 | 18.61 |
| | ContinuousSR | **27.23** | **25.16** | **24.01** | **23.21** | **22.61** | **21.97** | **21.42** | **21.13** | **19.83** | **18.86** |
| DIV2K | GSASR | 29.82 | 27.51 | 26.13 | 25.14 | 24.36 | 23.19 | 22.73 | 22.33 | 20.61 | 19.35 |
| | ContinuousSR | **29.95** | **27.59** | **26.19** | **25.19** | **24.42** | **23.28** | **22.83** | **22.44** | **20.75** | **19.51** |

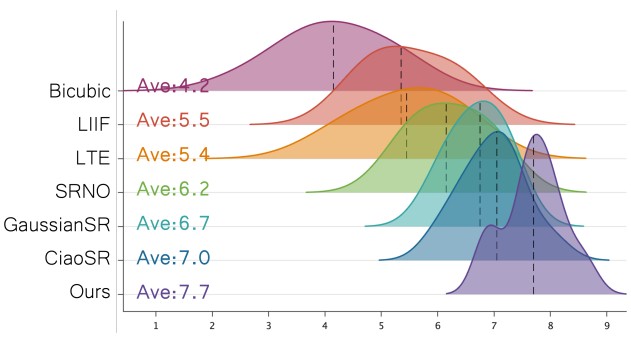

Figure 5: User study.

original three-dimensional setting.

**Ablation Study on number of $\mathcal{K}$.** Furthermore, we investigate the impact of the number of $\mathcal{K}$ on the network's performance. We define 100, 500, and 730 Gaussian covariances, and the results are presented in Table 13. It can be seen that as the number of covariances increases, the network's performance improves. However, beyond 730, no further performance gains are observed. Therefore, in this work, we set the number of covariances to 730.

**Ablation Study on $N$.** Additionally, we study the effect of the number of Gaussians $N$ on the network's performance. As described in the main text, we initialize one Gaussian kernel at the center of each LR pixel. We further explore the impact of introducing more Gaussian kernels per unit pixel, and the results are shown in Table 14. It can be observed that the best performance is achieved when the number of kernels is set to 4. Introducing too many kernels increases the optimization complexity, which does not lead to further performance gains. Therefore, we set the number of kernels per pixel to 4 in this work.

**Ablation Study on $P_{\mathbf{off}}$.** Moreover, we investigate the impact and motivation of the Adaptive Position Drifting (APD) module, which enhances spatial adaptability by allowing controlled positional shifts of Gaussian centers. This design enables the model to better capture local geometric variations and fine details in the image, while maintaining stability through an explicitly constrained offset range $[-A, A]$. To determine the optimal range, we set $A$ to 0.5, 1, and 2, and report the results in Table 15. It can be observed that setting the range to 1 achieves the best trade-off between representational flexibility and geometric accuracy, whereas both excessively large and small ranges lead to performance degradation.

**Ablation Study on the Color Gaussian Mapping (CGM) Module.** To provide a more detailed analysis of the Color Gaussian Mapping (CGM) module, we conducted an additional ablation study on its network depth using the Urban100 dataset. The CGM module is designed to learn the color parameters of Gaussian kernels and is implemented as a lightweight multilayer perceptron (MLP) that predicts RGB values directly from the encoded feature map. This structure enables the network to infer fine-grained color parameters conditioned on both local textures and global context, ensuring accurate and consistent color reconstruction at high scales. We varied the number of MLP layers from 4 to 8 and summarize the results in Table 17. The results show a clear improvement when increasing the depth from 4 to 6 layers, while the gain becomes marginal beyond six layers. Therefore, we adopt a six-layer MLP as a balanced configuration between reconstruction quality and computational efficiency.

Table 12: Ablation on Dim.

|  | PSNR | SSIM |
|---|---|---|
| 3 | 28.14 | 0.8273 |
| 256 | 28.17 | 0.8281 |
| 512 | 28.22 | 0.8292 |

Table 13: Number of $K$.

|  | PSNR | SSIM |
|---|---|---|
| 100 | 28.12 | 0.8277 |
| 500 | 28.19 | 0.8286 |
| 730 | 28.22 | 0.8292 |

Table 14: Number of $N$.

|  | PSNR | SSIM |
|---|---|---|
| 1 | 28.01 | 0.8254 |
| 4 | 28.22 | 0.8292 |
| 9 | 28.18 | 0.8284 |

Table 15: Ablation on $P_{\text{off}}$.

|  | PSNR | SSIM |
|---|---|---|
| 0.5 | 28.03 | 0.8259 |
| 1 | 28.22 | 0.8292 |
| 2 | 28.17 | 0.8283 |

Table 16: Performance comparison of SR and deraining methods under different scaling factors.

| Types | Methods | ×4 | ×4.5 | ×5 | ×5.5 | ×6 | ×6.5 | ×7 | ×7.5 | ×8 |
|---|---|---|---|---|---|---|---|---|---|---|
| ASSR+Derain | LIIF+DRSformer | 17.92 | 16.76 | 15.84 | 15.03 | 14.47 | 14.02 | 13.71 | 13.58 | 13.50 |
|  | DRSformer+LIIF | 20.14 | 19.87 | 19.76 | 19.53 | 19.39 | 19.21 | 19.12 | 18.98 | 18.93 |
|  | GaussianSR+DRSformer | 17.14 | 16.09 | 15.25 | 14.53 | 14.07 | 13.75 | 13.57 | 13.53 | 13.52 |
|  | DRSformer+GaussianSR | 20.12 | 19.86 | 19.75 | 19.52 | 19.39 | 19.21 | 19.13 | 18.98 | 18.94 |
|  | CiaoSR+DRSformer | 19.45 | 18.25 | 17.19 | 16.29 | 15.55 | 14.85 | 14.22 | 13.94 | 13.73 |
|  | DRSformer+CiaoSR | 19.96 | 19.73 | 19.65 | 19.48 | 19.27 | 19.15 | 19.07 | 18.89 | 18.87 |
| All in one | LIIF | 24.04 | 23.79 | 23.53 | 23.29 | 23.09 | 22.90 | 22.70 | 22.52 | 22.39 |
|  | GaussianSR | 24.04 | 23.76 | 23.51 | 23.28 | 23.08 | 22.89 | 22.69 | 22.52 | 22.38 |
|  | CiaoSR | 23.84 | 23.66 | 23.45 | 22.23 | 23.01 | 22.83 | 22.66 | 22.50 | 22.33 |
|  | Ours | 24.51 | 24.22 | 23.95 | 23.69 | 23.48 | 23.28 | 23.07 | 22.90 | 22.76 |

## A.6 ADDITIONAL VISUAL COMPARISON RESULTS

In this section, we present additional visual comparison results to further demonstrate the superiority of our proposed method, as shown in Figure 7, 8 and 9. It can be observed that our method achieves the best visual satisfaction in terms of detailed textures, while also preserving the highest level of detail fidelity, making it closest to the GT image.

## A.7 VISUALIZATION OF THE POSITION DISTRIBUTION

To demonstrate the superior adaptive perception capability of the proposed offset mechanism, which effectively introduces more Gaussian kernels in complex texture regions based on image content, we visualize the learned Gaussian position distribution. As shown in Figure 6, the proposed Adaptive Position Drifting adjusts the original initialization of the position distribution by adaptively perceiving the structural content of the image. The results reveal that regions with richer textures have higher densities of Gaussian kernels.

## A.8 ALGORITHM WORKFLOW

To clearly demonstrate the details of the proposed method, we design an algorithm workflow, as illustrated in Algorithm 1. This workflow describes the key steps from input to output, including feature encoding, color prediction, offset prediction, covariance estimation, and the final image reconstruction process.

## A.9 MORE EXPLORATION AND RESULTS

In Section 6 of the main text, we demonstrate the performance of our method in low-resolution and rainy scenarios. Here, we present comparisons across more scaling factors and with the two-stage ASSR+Derain methods DRSformer Chen et al. (2023b). The results are shown in Table 16. As

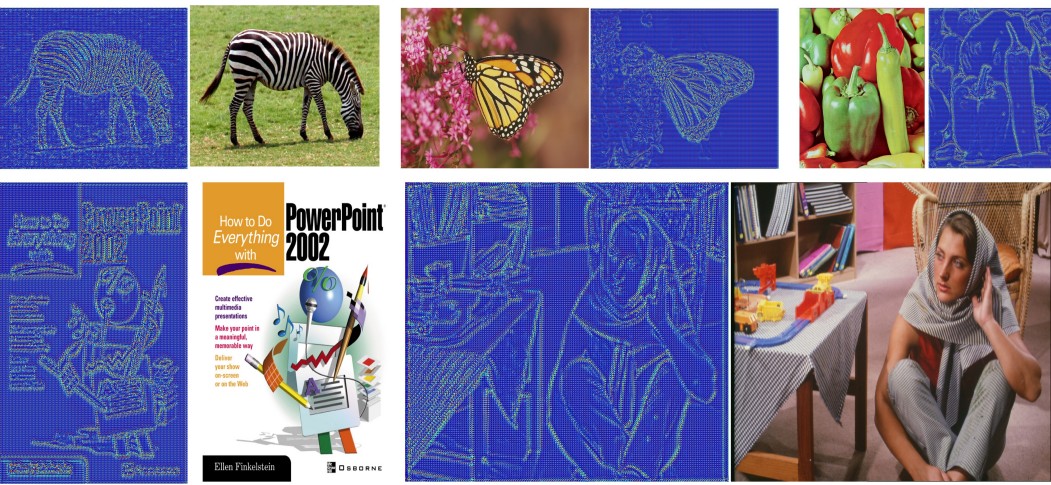

Figure 6: Visualization of the position distribution.

Table 17: Ablation Study on CGM Network Depth (Urban100 dataset).

| Number of MLP Layers | FLOPs (G) | Params (M) | PSNR (dB) |
|:---:|:---:|:---:|:---:|
| 4 | 3.39 | 0.20 | 28.12 |
| 5 | 13.11 | 0.79 | 28.19 |
| 6 | 15.80 | 0.96 | 28.22 |
| 7 | 22.27 | 1.35 | 28.22 |
| 8 | 26.59 | 1.62 | 28.23 |

shown in Table 16, our method consistently outperforms other methods across all scaling factors. For instance, at the ×4 scale, our method achieves a PSNR of 24.51, significantly higher than the best two-stage method, DRSformer Chen et al. (2023b)+LIIF (20.14). At the ×8 scale, our method achieves 22.76, outperforming DRSformer+LIIF (18.93). Compared to other "All in one" methods, our approach also achieves superior results, such as 23.95 at the ×5 scale, outperforming both GaussianSR (23.51) and CiaoSR (23.45). These results highlight the robustness, simplicity, and effectiveness of our method for super-resolution and deraining tasks across various scales.

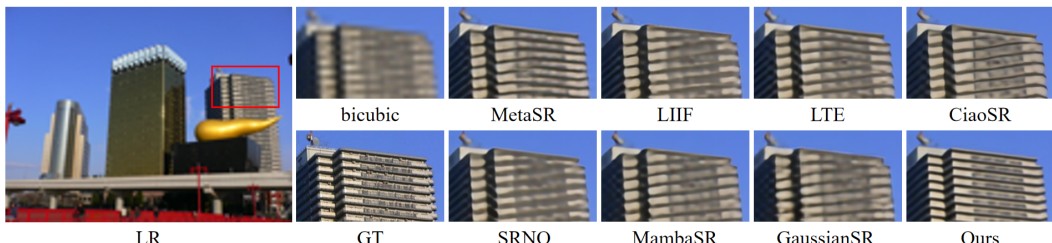

Figure 7: More qualitative comparison. The visual quality of our method outperforms existing methods. Please zoom in for a better view.

## A.10 LIMITATION AND FUTURE WORK

**Position Distribution.** In this paper, to address the difficulty of optimizing position parameters, we propose Adaptive Position Drifting, which leverages an offset mechanism to alleviate the optimization challenges and enhance the representational capacity of the model. However, assigning one or four Gaussian kernels to each LR pixel introduces some limitations. On the one hand , it leads to an overabundance of Gaussian kernels in low-frequency regions, resulting in resource wastage.

Table 18: Comparison with GaussianImage Zhang et al. (2024b).

|        | LIIF  | LTE   | CiaoSR | GaussianImage | Ours    |
|--------|-------|-------|--------|---------------|---------|
| Set5   | 32.73 | 32.81 | 32.84  | 28.01         | **33.24** |
| Set14  | 28.98 | 29.06 | 29.08  | 25.62         | **29.40** |

---

**Algorithm 1:** ContinuousSR.

---

**Input:** inp : (B, 3, $H_0$, $W_0$)
**Output:** image : (B, 3, H, W)
$F \leftarrow \text{Encoder(inp)}$;
$C \leftarrow \text{CGM}(F)$ ; // $[N \times 3]$
$\Delta x \leftarrow \text{APD}(F)$ ; // $[N \times 2]$
$\Sigma \leftarrow \text{DDCW}(F)$ ; // $[N \times 3]$
$\tilde{x} \leftarrow \text{Grid}(H, W) + \alpha \Delta x$
**for** $n \in [1, N]$ **do**
$\quad \lfloor \ (xys_n, \text{depth}_n, \text{radii}_n, \text{conic}_n) \leftarrow \text{Project}(\tilde{x}_n, \Sigma_n)$;
$\text{image} \leftarrow \text{Composite}(\{xys, \text{depth}, \text{radii}, \text{conic}, C\})$;

---

On the other hand, it increases the optimization difficulty significantly. To address these issues, we plan to explore the adaptive allocation of Gaussian kernels based on the texture complexity of image content in future work. This approach aims to dynamically assign an appropriate number of kernels to different regions, effectively mitigating the aforementioned problems.

**Introduce Generation Knowledge.** In addition, considering that arbitrary-scale super-resolution sometimes requires large upscaling factors (*e.g.*, ×16, ×32, *etc.*), it is challenging for the model to generate high-quality details solely relying on the input image and model knowledge. Therefore, in the future, we plan to incorporate more visual knowledge from diffusion models or semantic knowledge from large vision-language models to help the network generate finer details for high-magnification scenarios.

### A.11 USE OF LARGE LANGUAGE MODELS

Large Language Models (LLMs) were used only for grammar checking and text polishing. All research ideas, methods, and analyses are solely by the authors.

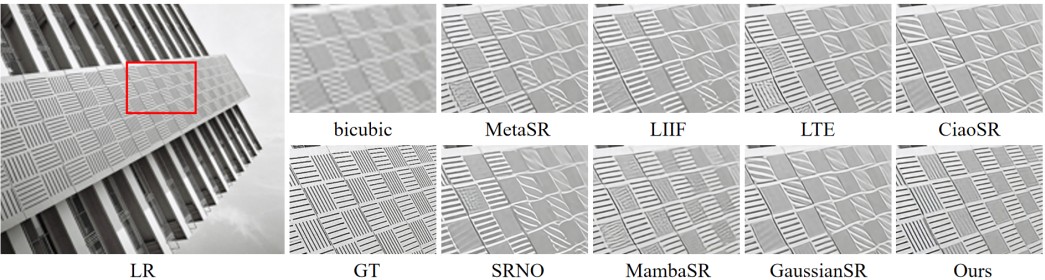

Figure 8: More qualitative comparison. The visual quality of our method outperforms existing methods. Please zoom in for a better view.

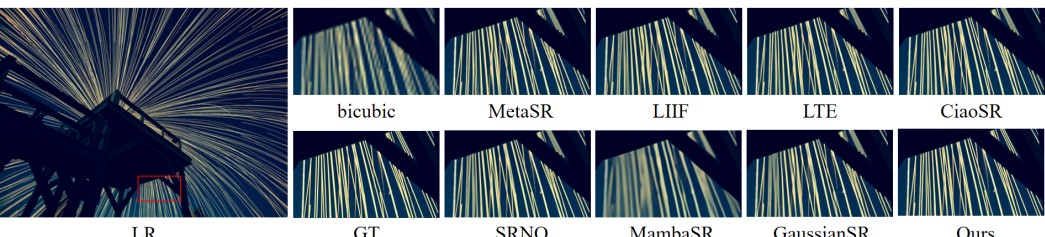

Figure 9: More qualitative comparison. The visual quality of our method outperforms existing methods. Please zoom in for a better view.

