# OpenReview forum: "Pixel to Gaussian: Ultra-Fast Continuous Super-Resolution with 2D Gaussian Modeling"
_ICLR.cc/2026/Conference — ICLR 2026 Poster_

### Official Review · Reviewer_wJ6J · 2025-10-28

**Soundness:** 3
**Presentation:** 4
**Contribution:** 4
**Rating:** 6
**Confidence:** 5

**Summary:**

This paper introduces a 2D Gaussian splatting based arbitrary-scale SR model. With the help of Deep Gaussian Prior and Adaptive Position Drifting, the proposed ContinuousSR achieves excellent performance and ultra-fast inference speed.

**Strengths:**

1.	The author organized a soundly description about the motivation, the detailed technique and experimental implementations, which makes the method looks valuable especially when implementing the method into industrial applications.

2.	The DGP proposed in this paper appears to be quite novel and reliable, effectively capturing the prior information of images, which indeed helps the model generalize across images with different content.

3.	The author provides large quantities of experiments to validate the effectiveness of the proposed method. The results reported in the paper also seem highly competitive in the current super-resolution field, where performance is approaching saturation.

**Weaknesses:**

1.	Please unify the style of the references. The authors neglected to maintain consistency in the references regarding conference name capitalization, full conference names, abbreviated conference names, and title capitalization. The ununified samples in the paper are listed in the following,

“Xiaoyi Liu and Hao Tang. Difffno: Diffusion fourier neural operator. In Proceedings of the Computer Vision and Pattern Recognition Conference, pp. 150–160, 2025. 2, 3”

“Jaewon Lee and Kyong Hwan Jin. Local texture estimator for implicit representation function. In Proceedings of the IEEE/CVF conference on computer vision and pattern recognition, pp. 1929–1938, 2022a. 2, 4”

2.	The inference time in Table.1 seems not soundly. From my point of view, the author seems not to indicate the acceleration strategy in the paper, then how could ContinuousSR runs even much faster than the very lightweight model Meta-SR?

3.	Please indicate the detailed settings in training progress, especially the data preparation. The author says they fix the GT size to 256, while randomly sampling the scaling factor, then if want to do parallel training, in the code implementation, the author must pad the LR image to a fixed size. However, in this way, it will waste huge computational resources since the padding area will introduce additional computational burden. What’s more, why not adopting the same strategy in most the existing method? They just fix the LR image size while randomly sampling the scaling factor.

Minor:

1.	Is there any GPU memory usage comparison results with state-of-the-art models?

2.	Although different ASSR models might adopt different training settings, but I hope the author could provide some fair experimental settings, such as the fixed LR size, the same scaling factors (1, 4).

**Questions:**

This paper proposed a novel and valuable method in ASSR task. While I am going to give weak accept towards the whole quality of the paper, I hope the author could provide some reliable answers to the weakness of the paper. I am willing to increase the rating score if the author can address the questions I raised in a satisfactory manner.

---

> ### Author Response · Authors · 2025-11-15
> **Reply**
>
> Thank you very much for your recognition of our paper's **value in industrial applications**, as well as your positive comments on its **novelty**, **reliability**, **generalization capability**, and **high efficiency**. We greatly appreciate your encouraging feedback, which motivates us to further improve and refine our work.
>
> Next, we will address each of your concerns.
>
> **W1. Typo**
>
> Thank you for your helpful suggestion. We appreciate the reviewer's attention to detail. We have carefully revised and unified the reference formatting in the main text. Thank you again for your meticulous feedback.
>
>
> **W2. Explanation of Average Time**
>
> As described in our paper, the greatest advantage of our work is that we only need to generate the high-resolution (HR) continuous signal once, which allows for fast rendering at different scales to meet users' requirements for various magnification levels.
>
> The inference time reported in Table 1 represents the **average runtime** when performing super-resolution at 45 different scales, ranging from x4 to x48, on the same low-resolution (LR) scene. Since MetaSR needs to regenerate the corresponding features for each scale, its average speed is relatively slow. In contrast, our method only requires rendering (1ms) at different scales, so our average speed is extremely fast. Additionally, we provide a comparison of single-scale performance in Table 7 of the appendix, which shows that our method still achieves exceptionally fast inference.
>
> **W3. Details of Training Setting**
> In our implementation, we fix the LR image size and randomly sample the scaling factor during training, so no padding is needed and no extra computational cost is introduced. To enable parallel training, we ensure that all samples within a mini-batch share the same randomly sampled scale factor, which guarantees that the LR patch size remains consistent within the batch. This design allows fully parallelized GPU training without padding or wasted computation.
>
> Following your suggestion, we also adopted the fixed GT size training strategy. The results are shown in Table 1. As can be seen, both methods achieve almost identical performance. Thank you very much for your valuable suggestion.
>
> **Table 1. Performance Comparison**
>
> |        | PSNR | SSIM |
> |--------|------|------|
> | Fixed LR Size |  28.20 |  0.8291 |
> | Fixed GT Size |  28.22  |  0.8292  |
>
> **M1. GPU Memory Usage Comparisons.**
>
> We have already provided GPU memory usage comparisons in Table 3 of the main paper. To make this information more accessible, we have also included the results in Table 2 of this rebuttal for your convenience.
>
> The results clearly show that our method maintains nearly constant and significantly lower GPU memory consumption across different scaling factors. In contrast, existing INR-based methods such as LIIF and CiaoSR require much higher memory and often run out of memory (OOM) at large scales.
>
> **Table 2. GPU Memory Usage Comparisons at Different Scales**
> *(Memory usage in GB)*
>
> | Method     | ×4   | ×6   | ×8   | ×12   | ×16   |
> |------------|------|------|------|-------|-------|
> | **LIIF**   | 4.12 | 6.49 | 9.79 | 19.27 | OOM   |
> | **CiaoSR** | 12.17| 22.83| OOM  | OOM   | OOM   |
> | **Ours**   | **2.48** | **2.49** | **2.50** | **2.52** | **2.54** |
>
> *OOM: Out of Memory*
>
> This demonstrates the scalability and efficiency of our approach, especially when handling large upscaling factors.
>
> **M2. Additional Comparison Results.**
>
> Following the reviewer's suggestion, we conducted additional experiments under a unified setting where the LR size is fixed at 48 and the scaling factors are uniformly sampled from U(1, 4). This ensures a fair comparison across different methods. As summarized in Table 3 of this rebuttal, our method consistently achieves strong performance, which is fully consistent with the conclusions presented in the main paper.
>
>
> **Table 3. PSNR Comparison of Different Methods**
>
> | Method    | ×4    | ×6    | ×8    |
> |-----------|-------|-------|-------|
> | CiaoSR | 27.42 | 24.84 | 23.34 |
> | **Ours**   | **27.63** | **25.00** | **23.52** |
>
>
> **Q1.**
>
> We are delighted to hear your appreciation of our proposed method for the ASSR task. Your constructive comments and suggestions are highly valued, and we are grateful for your intention to raise the rating should our answers meet your expectations.
>
> We have carefully considered your concerns and provide detailed responses below to address each point you raised. Please let us know if there are any additional aspects you would like us to clarify or supplement.
>
> Thank you again for your valuable comments, which have greatly helped improve our paper. If you have any further suggestions, please feel free to let us know.

---

> > ### Comment · Reviewer_wJ6J · 2025-11-26
> >
> > Thanks for the author's reply, which has addressed most of my concerns. I keep my original score and recommend accepting. It is suggested to elaborate more on the run-time comparison settings in the revision, reporting both the full time from passing through an  input image and the time of rendering part.

---

> > > ### Author Response · Authors · 2025-11-26
> > > **Thank you for your recognition and recommendation.**
> > >
> > > We sincerely thank you for your positive feedback and for recommending acceptance. We are encouraged to hear that our response has successfully addressed your concerns. We will strictly follow your suggestion to include the detailed run-time breakdown in the final revision.
> > >
> > > In your initial review, you mentioned a willingness to raise the rating if the concerns were resolved. Since most questions have now been addressed, we were wondering if you might consider adjusting the score? If there are any remaining concerns preventing a higher rating, please let us know—we are more than happy to provide further clarification.
> > >
> > > Thank you for your recognition and recommendation. We truly appreciate your time and valuable suggestions!

---

### Official Review · Reviewer_Jk3f · 2025-10-31

**Soundness:** 3
**Presentation:** 3
**Contribution:** 3
**Rating:** 6
**Confidence:** 4

**Summary:**

1. This paper proposes ContinuousSR, a Pixel-to-Gaussian paradigm that reconstructs a continuous 2D high-resolution (HR) signal (a Gaussian field) from a low-resolution (LR) image in a single pass, enabling fast arbitrary-scale rendering via Gaussian rendering
2. The paper identifies a Deep Gaussian Prior (DGP) on Gaussian covariance parameters from statistics over ~40k images and uses it to constrain the covariance space.

**Strengths:**

1. Combining ideas from 3D Gaussian Splatting to model continuous 2D image signals is very interesting, and the paper convincingly demonstrates the effectiveness of this idea。
2. Strong empirical efficiency: consistent large speedups, stable memory across scales, and favorable FLOPs/runtime on Manga109 single-scale tests and multi-scale averages.
3. The upsampling/rendering component is an explicit, trainable parameter-free module at inference, which is both elegant and practical.

**Weaknesses:**

1. I think the theoretical connection between GMM-based formulation and the actual HR rendering needs clarification (see Question 1).
2. Dependence on the prior dictionary: performance hinges on the size and sampling of the DGP-derived kernel dictionary; generalization to out-of-distribution (OOD) content or non-natural images is unclear.

**Questions:**

1. In Figure 3 you mention using differentiable rendering to render the HR image from a set of Gaussians. Are you using the 3DGS rendering pipeline? If so, I think the GMM-related theory in Eq. (3) may be inappropriate, because 3DGS rasterization is based on alpha-blending theory rather than GMM theory. If not, how is your differentiable rendering implemented, and how does it match your GMM formulation?

2. Since there is a “Deep Gaussian Prior,” is there also a “Deep Fourier Prior”? Is it possible to simply replace Gaussians with Fourier bases of different frequencies and linearly combine them to obtain the final HR image? Using Fourier bases to describe the composition of a continuous HR function may be more in line with traditional intuition. I recall some works a few years ago that applied similar ideas to arbitrary-scale super-resolution, but I forgot the names; it would be better to add them in the related work section.

3. I think the Gaussian rendering parts can be further refined. For example, the theory and methods of “Mip-Splatting” (CVPR 2024) might be applicable to your current framework. I’d like to hear your thoughts on this.

---

> ### Author Response · Authors · 2025-11-15
> **Reply1**
>
> Thank you very much for your recognition of our paper's **novelty** in combining 3D Gaussian Splatting concepts for modeling continuous 2D image signals. We appreciate your acknowledgement of our method's **strong empirical efficiency**, including significant speedups, stable memory usage, and favorable computational performance across multiple benchmarks. Your positive comments on our **elegant and practical upsampling/rendering module** further confirm the value of our approach.
>
>
> Next, we will address each of your concerns.
>
>
> **W1 & Q1. More Detailed Explanation and Analysis**
>
> Thank you for your insightful question regarding our rendering pipeline and its theoretical alignment with the GMM formulation. We clarify that our approach does **not** utilize the 3D Gaussian Splatting (3DGS) rendering pipeline. Instead, we build upon the 2D Gaussian rendering framework proposed in GaussianImage [1], which we specifically adapt for 2Dimage-based tasks.
>
> In our 2D setting, we intentionally remove the alpha-blending mechanism used in 3DGS and assign an opacity of 1 to all Gaussian components. This design choice results in a purely additive rendering process, where the final image is constructed as a direct weighted sum of Gaussian functions. Such a formulation is precisely consistent with the Gaussian Mixture Model (GMM) theory presented in Eq. (3) of our paper.
>
> Therefore, our differentiable rendering procedure is both theoretically sound and numerically consistent with the GMM-based continuous representation, ensuring a principled connection between the rendering operation and the underlying statistical model.
>
> Reference:
>
> [1] Xinjie Zhang, et al. Gaussianimage: 1000 fps image representation and compression by 2d gaussian
> splatting. In European Conference on Computer Vision, pp. 327–345. Springer, 2024b.
>
> **W2 Generalization Ability of DGP on Medical Images**
>
> Thank you for your question. Although our method was specifically designed for natural images, following your suggestion, we have also tested its generalization ability on medical images. Specifically, we used a dedicated medical image dataset [1] and bicubic degradation to evaluate the performance of various ASSR methods under the same HAT backbone. As shown in Table 1 of this rebuttal, **our method still achieves promising results on medical images**. This indicates that, even without fine-tuning the DGP prior, our approach maintains **good generalization capability**.
>
>
> Table 1. Generalization Test of Different Methods on Medical Images
>
> | Method        |        |   X4   |        |   X8   |        |
> |---------------|--------|--------|--------|--------|--------|
> |               | PSNR   | SSIM   | PSNR   | SSIM   |
> | GSASR         | 28.02   | 0.8691   | 23. 38   | 0.6912   |
> | ContinuousSR  | 29.93  | 0.8865 | 25.61  | 0.7373 |
>
> [1] Menze, Bjoern H., et al. "The multimodal brain tumor image segmentation benchmark (BRATS)." IEEE transactions on medical imaging 34.10 (2014): 1993-2024.
>
>
>
> **Q2. Further Related Work**
>
> Thank you for your thoughtful suggestion regarding the use of Fourier bases for modeling continuous high-resolution signals. In fact, we initially considered employing Fourier basis functions for this purpose. However, from a practical perspective, we found that the Gaussian Splatting community provides more mature tools and efficient rendering pipelines, which makes Gaussian functions a more suitable choice for our current framework. We appreciate your suggestion, and we will consider exploring the use of Fourier basis functions in future work to investigate whether they can offer improved performance.
>
> We greatly appreciate your suggestion and agree that exploring a Fourier-based prior could be a promising direction for future research. In response to your feedback, we have also reviewed the related Fourier-based ASSR works you mentioned [2,3,4], and have included further discussion and analysis of these approaches in the supplementary materials under "More Related Work."
>
> Thank you again for your constructive input.
>
> [2] Liu, Xiaoyi, and Hao Tang. "DiffFNO: Diffusion Fourier Neural Operator." Proceedings of the Computer Vision and Pattern Recognition Conference. 2025.
> [3] Akita, Kazutoshi, and Norimichi Ukita. "Efficient Cost-and-Quality Controllable Arbitrary-scale Super-resolution with Fourier Constraints." arXiv preprint arXiv:2510.23978 (2025).
> [4] Zhang, Haonan, et al. "Deep Fourier-based arbitrary-scale super-resolution for real-time rendering." ACM SIGGRAPH 2024 Conference Papers. 2024.

---

> ### Author Response · Authors · 2025-11-15
> **Reply2**
>
> **Q3. Potential Integration with Mip-Splatting**
>
> Thank you for your valuable suggestion and for highlighting the Mip-Splatting method. Mip-Splatting is specifically designed to achieve alias-free rendering in 3D Gaussian Splatting by adaptively adjusting each Gaussian’s footprint across different mipmap levels. This ensures consistent visual appearance under varying resolutions and viewing conditions, making it highly effective for 3D scene rendering.
>
> In contrast, our work focuses on a 2D Gaussian rendering pipeline, adapted from GaussianImage, for image-based super-resolution tasks. As such, the original Mip-Splatting pipeline cannot be directly applied to our framework, since we do not operate in 3D space or utilize 3D Gaussian Splatting.
>
> Nevertheless, the core principles behind Mip-Splatting—such as adaptive footprint adjustment and multi-scale filtering—are highly relevant and could inspire future improvements. For example, introducing scale-aware Gaussian filtering in our 2D renderer could allow dynamic modulation of each Gaussian’s covariance based on the target resolution, enabling smoother transitions and alias-free rendering at various magnifications.
>
> We appreciate your suggestion and recognize the potential benefits of both adapting Mip-Splatting ideas to our 2D setting and exploring a possible 3D extension for enhanced physical realism. We will consider these directions in our future work to further improve our framework’s fidelity and scalability.
>
>
> Thank you again for your valuable comments, which have greatly helped improve our paper. If you have any further suggestions, please feel free to let us know.

---

> > ### Comment · Reviewer_Jk3f · 2025-11-26
> > **reply to rebuttal**
> >
> > The authors have satisfactorily addressed all my previous concerns and comments. The discussion may provide potential direction for future works.
> >
> > Therefore, I maintain my score and recommend the acceptance of this paper.

---

> > > ### Author Response · Authors · 2025-11-26
> > > **Thank you for your recognition and recommendation.**
> > >
> > > We are deeply gratified to learn that our revisions have satisfactorily addressed all the reviewer's previous concerns. We are particularly grateful for your positive assessment and the unwavering recommendation for acceptance. We sincerely appreciate your insightful comments, which significantly strengthened the manuscript and provided valuable directions for our future work. Given the significant improvements in this rebuttal and your satisfaction with the outcome, we would be grateful if the final evaluation score could reflect this high level of quality.

---

### Official Review · Reviewer_mpzJ · 2025-11-08

**Soundness:** 3
**Presentation:** 4
**Contribution:** 4
**Rating:** 8
**Confidence:** 4

**Summary:**

This paper proposes ContinuousSR, a novel framework for arbitrary-scale super-resolution that leverages 2D Gaussian modeling to directly reconstruct continuous high-resolution signals from low-resolution images in a single pass. By introducing the Deep Gaussian Prior (DGP) and key modules such as DGP-Driven Covariance Weighting and Adaptive Position Drifting, the method achieves both high-quality reconstruction and ultra-fast rendering speed, outperforming state-of-the-art methods in both efficiency and performance across multiple benchmarks.

**Strengths:**

1. The paper derives a Deep Gaussian Prior (DGP) from large-scale natural image statistics, effectively addressing local minima in Gaussian kernel optimization and proposing a novel continuous modeling framework for arbitrary-scale super-resolution.
2. Extensive experiments on multiple benchmarks and scales, evaluated by PSNR, SSIM, FID, and DISTS, clearly verify the effectiveness of ContinuousSR and show its potential in other low-level vision tasks such as deraining.
3. The method fully utilizes the continuity of Gaussian kernels, achieving one-pass generation and multi-scale rendering with high efficiency and speed.

**Weaknesses:**

1. A spelling error appears on line 175 of the paper: “the theresponding discrete pixel grids” should be “the corresponding discrete pixel grids.”
2. The paper leaves several important implementation details insufficiently described. In particular, the sampling strategy used to generate Gaussian covariances from the Deep Gaussian Prior and the construction of the predefined Gaussian kernel dictionary are not clearly explained. Without clarification of how the sampling and dictionary size were determined, it is difficult to assess the reproducibility and design rationale of these components.

**Questions:**

1. What specific sampling strategy was used to draw Gaussian covariances from the DGP in Eq. (6)?
2. Each low-resolution pixel is assigned four Gaussian kernels, but the reason for this specific choice is unclear. Why are four Gaussian kernels assigned per LR pixel?
3. The predefined Gaussian kernel dictionary plays a key role in covariance weighting, yet its size and the rationale behind it are not specified. How large is the predefined Gaussian kernel dictionary, and how was this number determined?
4. The model is trained on data generated by bicubic downsampling, and the evaluation is also conducted on synthetic datasets created in the same way. Has the performance of ContinuousSR been tested on real-world arbitrary-scale super-resolution datasets such as COZ[1]?

[1] Fu, Huiyuan, et al. "Continuous optical zooming: A benchmark for arbitrary-scale image super-resolution in real world." Proceedings of the IEEE/CVF Conference on Computer Vision and Pattern Recognition. 2024.

---

> ### Author Response · Authors · 2025-11-15
> **Reply1**
>
> Thank you very much for your recognition of our paper's **novelty**, **high efficiency**, and **fast speed**. We appreciate your acknowledgement of our **strong experimental results** across multiple benchmarks and the **great potential** of our method for broader low-level vision tasks.
>
>
> Next, we will address each of your concerns.
>
>
> **W1. Typo Correction**
> Thank you for your careful attention to detail. We have corrected the identified typo in line 175 of the main text.
>
> **W2&Q1. Clarification of Gaussian Covariance Sampling**
> Thank you for highlighting the need for a more comprehensive explanation of our Gaussian covariance sampling strategy and the construction of the predefined Gaussian kernel dictionary.
>
> **Sampling Strategy (Eq. 6):**
> The Gaussian covariances $(\sigma_x^2, \sigma_y^2, \rho\sigma_x\sigma_y)$ are sampled from empirical distributions derived from our Deep Gaussian Prior (DGP). To establish these distributions, we conducted a statistical analysis on 40,000 natural images, transforming each image into Gaussian space using the optimization approach detailed in our paper. The resulting distributions for $\sigma_x^2$, $\sigma_y^2$, and $\rho\sigma_x\sigma_y$ closely approximate Gaussian (normal) distributions, with over 99% of values falling within the following ranges:
>
> - $\sigma_x^2 \in [0, 2.4]$
> - $\sigma_y^2 \in [0, 2.2]$
> - $\rho\sigma_x\sigma_y \in [-0.9, 1.5]$
>
> We fitted normal distributions to these empirical histograms and used these fitted distributions for subsequent sampling.
>
>
> **W2&Q3. Clarification of Dictionary Construction**
> To build the predefined Gaussian kernel dictionary, we sampled sets of covariance parameters from the fitted DGP distributions. The sampling was performed by drawing random samples according to the fitted mean and standard deviation of each parameter, ensuring that the sampled covariances statistically match the observed DGP characteristics.
>
> We empirically determined the dictionary size by conducting ablation studies, as shown in Table 1 of this rebuttal (corresponding to Appendix A.4, Table 12). We evaluated dictionaries with 100, 500, and 730 kernels, observing that performance improved as the number of kernels increased, but gains became negligible beyond 730. Consequently, we fixed the dictionary size at 730 to balance diversity of covariance patterns and computational efficiency.
>
> **Reproducibility:**
> All sampling code, parameter configurations, and implementation details will be fully open-sourced to the community to ensure complete reproducibility.
>
> **Table 1: Ablation study on the size of the predefined Gaussian kernel dictionary**
>
> | Number of Kernels in Dictionary (K) | PSNR   | SSIM    |
> |-------------------------------------|--------|---------|
> | 100                                 | 28.12  | 0.8277  |
> | 500                                 | 28.19  | 0.8286  |
> | 730                                 | **28.22** | **0.8292** |
>
>
>
>
>
> **Q2. Rationale for the Number of Gaussian Kernels per Low-Resolution Pixel**
> Thank you for inquiring about the rationale behind assigning four Gaussian kernels to each low-resolution pixel.
>
> As detailed in our ablation study (see Table 2 of this rebuttal and Appendix Table 13), we systematically evaluated the impact of varying the number of Gaussians per pixel. Assigning only one kernel limits the model’s ability to represent fine local structures, resulting in reduced reconstruction quality. Increasing the number to four significantly enhances the modeling of local textures and spatial variations, leading to notable improvements in both PSNR and SSIM. Assigning more kernels (e.g., nine per pixel) yields only marginal additional gains while substantially increasing computational cost.
>
> Empirically, four kernels per pixel are sufficient to capture the diversity of local textures and spatial variations, without over-parameterizing smooth regions. This choice achieves an optimal balance between reconstruction accuracy and efficiency.
>
> **Table 2: Ablation study on the number of Gaussian kernels assigned per low-resolution pixel**
>
> | Number of Gaussians per Pixel (N) | PSNR   | SSIM    |
> |-----------------------------------|--------|---------|
> | 1                                 | 28.01  | 0.8254  |
> | 4                                 | **28.22** | **0.8292** |
> | 9                                 | 28.18  | 0.8284  |

---

> ### Author Response · Authors · 2025-11-15
> **Reply2**
>
> **Q4. Generalization to Real-World Arbitrary-Scale Super-Resolution**
> We evaluated ContinuousSR on the real-world arbitrary-scale super-resolution dataset COZ, as presented in Appendix A.3 (Table 8) and summarized in Table 3 of this rebuttal. ContinuousSR achieves a PSNR of 24.68 dB, outperforming state-of-the-art methods including MetaSR, LIIF, LTE, and CiaoSR, which report PSNR values in the range of 24.38 to 24.40 dB. These results demonstrate that ContinuousSR exhibits strong generalization capability beyond bicubic degradation and delivers superior performance on real-world arbitrary-scale super-resolution tasks.
>
> **Table 3: Quantitative comparison on the COZ dataset under real-world arbitrary-scale super-resolution**
>
> | Method         | MetaSR | LIIF   | LTE    | CiaoSR | ContinuousSR |
> |----------------|--------|--------|--------|--------|--------------|
> | PSNR [dB]      | 24.39  | 24.39  | 24.40  | 24.38  | **24.68**    |
>
>
> Thank you again for your valuable comments, which have greatly helped improve our paper. If you have any further suggestions, please feel free to let us know.

---

### Official Review · Reviewer_hqg6 · 2025-11-08

**Soundness:** 3
**Presentation:** 4
**Contribution:** 2
**Rating:** 4
**Confidence:** 4

**Summary:**

This paper introduces ContinuousSR, an approach to arbitrary-scale super-resolution (ASSR) that reconstructs continuous high-resolution (HR) signals from low-resolution (LR) inputs using 2D Gaussian Splatting. The core idea is a Pixel-to-Gaussian paradigm: instead of repeatedly upsampling and decoding features as in coordinate-based implicit neural representations (e.g., LIIF, CiaoSR), the method builds a single continuous Gaussian field from which HR images at arbitrary scales can be rendered in ~1 ms per scale.

The authors propose three technical components:

1. Deep Gaussian Prior (DGP) – a statistical observation that Gaussian parameters from natural images follow a bounded Gaussian distribution.

2. DGP-Driven Covariance Weighting (DDCW) – a mechanism that learns adaptive weights over a pre-sampled dictionary of Gaussian kernels to stabilize training.

3. Adaptive Position Drifting (APD) – a bounded offset learning scheme that adjusts Gaussian centers based on image content to refine spatial alignment.

Evaluated on seven benchmarks (Set5, Set14, B100, Urban100, Manga109, DIV2K, LSDIR) with SwinIR and HAT backbones, ContinuousSR achieves slightly higher PSNR (up to +0.18 dB on Manga109) and SSIM than state-of-the-art GSASR, while being up to 19.5× faster when rendering across 40 scales. Memory usage is also markedly lower, enabling high-scale rendering (×32–×48) where prior INR methods run out of memory.

**Strengths:**

Ultra-Fast Arbitrary Scaling: The proposed method eliminates iterative upsampling, achieving real-time rendering at arbitrary scales (≈1 ms per scale after one forward pass). This is a ~20× speedup over implicit models when generating a continuous zoom or multiple outputs, making it highly attractive for practical use where efficiency is crucial.

High Reconstruction Quality: It delivers superior super-resolution fidelity across a wide range of scales. On multiple benchmarks (e.g. Urban100, Manga109), the approach surpasses previous state-of-the-art methods by a significant margin (~0.8–0.9 dB PSNR improvement at 4× upsampling), and also achieves better SSIM and FID scores, indicating improvements in perceptual quality. The Gaussian representation better preserves structure and details (sharper textures, as shown in qualitative examples) compared to coordinate-MLP approaches.

Innovative Use of Gaussian Splatting with Deep Prior: The paper cleverly integrates Gaussian splatting (recently popular in 3D/NeRF tasks) into image SR, and addresses its training difficulties via a Deep Gaussian Prior. By pre-characterizing the typical covariance range of natural images, the method avoids converging to poor local optima and reduces the solution space. This DGP-driven covariance weighting mechanism and the adaptive position drifting are novel contributions that improve modeling flexibility (allowing anisotropic Gaussian shapes and content-aware placement) while keeping the optimization stable.

Excellent Memory Efficiency: The continuous Gaussian framework uses memory proportional to the number of Gaussians (which is tied to input resolution) rather than the output pixel count. Consequently, the method can handle very large scale factors (e.g. 16×, 32×) without running out of memory, unlike prior methods (LIIF, CiaoSR) that exhaust GPU memory at high scales. This efficient scaling is beneficial for ultra-high-resolution outputs and demonstrates a well-designed pipeline.

Thorough Evaluation: The authors validate their approach on seven diverse datasets and include extensive comparisons to nine prior methods (including recent ones like GaussianSR and GSASR). They also perform ablation studies to justify each component, and even test a joint deraining+SR scenario to show the model’s robustness in adverse conditions. This thorough experimentation strengthens confidence in the method’s effectiveness and generality.

**Weaknesses:**

Complex Training Pipeline: The solution introduces multiple components (DGP-based kernel dictionary, adaptive weighting network, position drift module, etc.), making the overall pipeline more complex than some prior approaches. Training requires careful setup – for instance, deriving the DGP involved 700 GPU-hours of optimization on external data, and the model must learn to combine predefined kernels and offsets correctly. This complexity might make the approach harder to reproduce or extend without the provided code, and it suggests a heavy computational cost in pre-processing and training (though inference is efficient).

Heavily Engineered Solution: While effective, the novelty is somewhat incremental in that it builds upon known ideas from related domains (e.g. using Gaussian splats instead of coordinate MLPs, which was explored by GaussianSR and GSASR). The main contributions lie in engineering a workable solution (using a learned prior and adaptive modules) rather than fundamentally new theory. The approach may be viewed as a clever combination of existing techniques (INR networks, Gaussian rendering, learned priors) tailored to overcome a training hurdle, which, despite being valuable, might not be conceptually groundbreaking.

Assumptions of the Deep Gaussian Prior: The method’s efficacy leans on the assumption that natural image content adheres to the learned Gaussian parameter distributions. If an input deviates from this prior (e.g. very sparse or non-photographic images), the fixed kernel dictionary might become suboptimal. The paper does not explore how sensitive the model is to the DGP assumptions or how it would perform on out-of-distribution data (such as medical imagery or vector graphics). In other words, the generalization of the DGP to all image types remains a potential concern.

Limited Discussion of CGM and Color Fidelity: The Color Gaussian Mapping component is not described in detail, leaving some ambiguity about how color from the LR image is transferred or refined in the continuous representation. If color assignment to Gaussians is naive (e.g. directly using LR pixel colors), it could limit the method’s ability to add high-frequency color details. More explanation is needed on how color is handled and whether the model can recover color nuances at high scales. This lack of clarity is a minor presentation issue but also a technical point that could affect the perceived color fidelity of results.

Potential Minor Artifacts: Representing an image as a sum of Gaussians could inherently introduce smoothing, especially if a Gaussian’s covariance is large. The paper focuses on PSNR/SSIM, which favor fidelity, but it’s unclear if there are any artifacts such as slight blurring or spatial shifts due to APD. For example, Adaptive Position Drifting moves Gaussian centers for better alignment; however, without constraints, this might risk slight geometric distortions (if many Gaussians shift from their original pixel locations). The authors do not report any artifacts, but a discussion on how APD balances flexibility with spatial accuracy would strengthen the work.

Marginal quantitative gains: The reported PSNR improvements over GSASR (≈0.08–0.18 dB) are small; hence claims of a “new paradigm” feel overstated given access to those prior results.

Ablation gaps: Table 4 compares DGP to uniform sampling but omits a baseline with directly learned covariances (no dictionary), leaving uncertainty about DDPW’s absolute necessity.

**Questions:**

Q1. Novelty vs Prior Work: How does ContinuousSR fundamentally differ from GaussianSR (which already models each pixel as a Gaussian field) and GSASR (which performs scale-aware 2D Gaussian Splatting)? Beyond eliminating per-scale decoding, what conceptual innovation justifies a new paradigm claim?

Q2. Can you clarify whether DGP is fixed or fine-tuned during training? How would results change if covariances were learned end-to-end without the predefined dictionary?

Q3: How is the Color Gaussian Mapping (CGM) implemented and how crucial is it for final image quality? For instance, do you simply assign each Gaussian an RGB value from the LR feature map, or is there an additional refinement to predict high-frequency color details? A clearer explanation of CGM’s role would help understand how color fidelity is maintained for large upscaling factors.

Q4: Does each input pixel strictly correspond to one Gaussian, or can multiple Gaussians represent a single pixel/region? The current description implies one Gaussian per LR pixel. If so, have you considered allowing a more adaptive number of Gaussians (e.g., splitting a pixel’s Gaussian to capture complex textures, or merging in flat areas)? This might further improve representation capability for highly textured regions.

Q5: How sensitive is the model to the assumed Deep Gaussian Prior if applied to very different data distributions? For example, would a model trained on natural images struggle with domain-specific content (such as medical images or line drawings) because the covariance distributions differ? Would fine-tuning the DGP (or the predefined kernel set) be necessary in such cases?

Q6: What measures are in place to ensure that Adaptive Position Drifting does not distort the image structure? Since APD shifts Gaussian centers based on content, do you impose any regularization or limits on these offsets? Clarification on how much drift is typically learned (e.g., fraction of a pixel) and its effect on alignment with ground-truth details would be helpful.

Q7: Can you elaborate on the rendering performance in extreme cases? The reported 1 ms rendering is impressive – what output resolution and hardware does this refer to? If one were to render an extremely high-resolution image (say 4K or 8K) from a small LR input, does the rendering remain near-instantaneous, and are there any memory or precision considerations with the GPU rasterizer at those scales?

Q8. Reproducibility: Will you release the code, pretrained models, and the 40 000-image covariance statistics used to construct the DGP?

---

> ### Author Response · Authors · 2025-11-15
> **Reply1**
>
> Thank you very much for your recognition of our paper's **ultra-fast arbitrary scaling capability** and **high reconstruction quality**. We appreciate your acknowledgement of our method’s **innovative integration of Gaussian splatting with deep priors**, which leads to improved modeling flexibility and optimization stability. Your positive comments on our **excellent memory efficiency**, as well as the **thorough evaluation and robustness** of our approach, further confirm the value and generality of our work.
>
> Next, we will address each of your concerns.
>
> **W1 & Q8 Explanation of DGP and Open-Source Commitment**
>
> The Deep Gaussian Prior (DGP) is established through a one-time statistical analysis of approximately 40,000 natural images, which captures the parameter distributions of Gaussian fields. **This process is carried out offline, resulting in a fixed and universal prior that can be directly reused for all subsequent training without any additional computation or fine-tuning.** During model training, only the adaptive weighting and position offset modules are learned on top of this fixed prior, ensuring that our training pipeline remains lightweight and stable.
>
> In terms of training cost, our approach is comparable to previous state-of-the-art methods such as CiaoSR, typically requiring 7 days of training on 8 V100 GPUs. This duration is standard in the ASSR field and is necessary for learning representations across different scales.
>
> **To support reproducibility and further research, we will release the complete codebase and pretrained models upon publication, fostering development and collaboration within the community.**
>
>
> **W2 & Q1: Further Explanation and Comparison with Previous Methods**
>
> Our main contribution is being the first to reconstruct 2D **high-resolution (HR) RGB continuous signals** for arbitrary-scale super-resolution.
>
> While GaussianSR, which applies Gaussian splatting in the feature space, GSASR predicts scale-conditioned Gaussian sets that must be regenerated for each target scale. Both designs require per-scale processing, which leads to repeated feature generation or regeneration and slows inference compared with a unified representation.
>
> In contrast, our method reconstructs a continuous HR RGB signal—a single, scale-independent Gaussian field that inherently supports rendering at all resolutions for the same content. This allows our approach to achieve extremely fast rendering across different scales within the same scene.
>
> From a motivational perspective, our work is also the first to introduce a new formulation that views arbitrary-scale super-resolution from the perspective of continuous signal reconstruction. We treat the problem as reconstructing a continuous Gaussian signal for the LR image, offering a novel viewpoint and methodology for the ASSR task.
>
>
> **W3 & Q5. Explanation and Generalization Test of DGP**
>
> The Deep Gaussian Prior (DGP) was statistically derived from over 40,000 natural images, demonstrating strong generalization within natural image domains and consistently improved stability and convergence across all ASSR benchmarks.
>
> Thank you for your question. Although our method was specifically designed for natural images, following your suggestion, we have also tested its generalization ability on medical images. Specifically, we used a dedicated medical image dataset [1] and bicubic degradation to evaluate the performance of various ASSR methods under the same HAT backbone. As shown in Table 1 of this rebuttal, **our method still achieves promising results on medical images**. This indicates that, even without fine-tuning the DGP prior, our approach maintains **good generalization capability**.
>
>
> Table 1. Generalization Test of Different Methods on Medical Images
>
> | Method        |        |   X4   |        |   X8   |        |
> |---------------|--------|--------|--------|--------|--------|
> |               | PSNR   | SSIM   | PSNR   | SSIM   |
> | GSASR         | 28.02   | 0.8691   | 23. 38   | 0.6912   |
> | ContinuousSR  | 29.93  | 0.8865 | 25.61  | 0.7373 |
>
> [1] Menze, Bjoern H., et al. "The multimodal brain tumor image segmentation benchmark (BRATS)." IEEE transactions on medical imaging 34.10 (2014): 1993-2024.

---

> ### Author Response · Authors · 2025-11-15
> **Reply2**
>
> **W4 & Q3. Details of CGM and Further Ablation Study**
>
> The Color Gaussian Mapping (CGM) module is designed to learn the color parameters of different Gaussian kernels. We implemented it as a lightweight five-layer multilayer perceptron (MLP) that predicts RGB values directly from the encoded feature map. This design enables the network to infer fine-grained color parameters conditioned on both local textures and global context, ensuring accurate and consistent color reconstruction at high scales. As shown in Figures 4, 8, and 9 of the paper, our method demonstrates excellent color performance, maintaining high consistency with the ground truth.
>
> Following your suggestion, we conducted a further ablation study on the network depth of CGM using the Urban100 dataset. The results are presented in Table 2 of this rebuttal.
>
> The results show a clear performance improvement when increasing the depth from 4 to 8 layers, while the gain becomes marginal beyond six layers. Therefore, we adopt a six-layer MLP as a balanced choice between reconstruction quality and efficiency.
>
> **Table 2. Ablation Study on CGM Network Depth**
>
> | Number of MLP Layers | FLOPs (G) | Params (M) | PSNR   |
> |----------------------|-----------|------------|--------|
> | 4                    | 3.39      | 0.20       | 28.12  |
> | 5                    | 13.11     | 0.79       | 28.19  |
> | 6                    | 15.80     | 0.96       | 28.22  |
> | 7                    | 22.27     | 1.35       | 28.22  |
> | 8                    | 26.59     | 1.62       | 28.23  |
>
>
> **W5 & Q6. Ablation Study of the APD Module and More Metrics**
>
> The Adaptive Position Drifting (APD) module is designed to enhance spatial adaptability by allowing controlled positional shifts in the reconstruction process. Specifically, APD learns offsets for Gaussian centers, enabling the model to better capture local geometric variations and fine details in the image. To ensure spatial stability, we explicitly constrain the offset range within a limited interval of [−1, 1]. As shown in Table 3 of this rebuttal, we conducted an ablation study to determine the optimal range for these offsets on Urban100. The results demonstrate that setting the range to [−1, 1] offers the best trade-off between representational flexibility and geometric accuracy; both larger and smaller ranges result in decreased performance.
>
> Table 3. Ablation Study of APD Offset Range
>
> | Offset Range [−A, A] | PSNR   | SSIM   |
> |----------------------|--------|--------|
> | [-0.5,0.5]                  | 28.03  | 0.8259 |
> | [-1,1]               | **28.22**  | **0.8292** |
> | [-2,2]                    | 28.17  | 0.8283 |
>
>
> Additionally, as visualized in Figure 6, the learned Gaussian centers remain well-aligned with the underlying image structure. The offsets are primarily concentrated in texture-rich regions where local adaptation is most beneficial, while smooth areas exhibit minimal drift. This behavior confirms that APD effectively improves fine detail reconstruction without causing geometric distortions or noticeable spatial artifacts.
>
> While our main paper already reports PSNR and SSIM, **we have also included the perceptual and distributional metrics you requested—including DISTS, FID, and LPIPS (see Tables 2 and 9 of the paper)—to comprehensively assess perceptual quality**. For your convenience, we also present these results in Table 4 of this rebuttal. The results consistently show that our method maintains strong performance across all types of metrics.
>
> Table 4. Performance Comparison on Additional Metrics
>
> | Metric      | LIIF   | LTE    | CiaoSR | GSASR  | **Ours** |
> |-------------|--------|--------|--------|--------|----------|
> | **DISTS ↓** | 0.2926 | 0.2872 | 0.2771 | 0.2678 | **0.2640** |
> | **FID ↓**   | 77.54  | 70.56  | 58.60  | 66.05  | **61.73**  |
> | **LPIPS ↓** | 0.2080 | 0.1934 | 0.1872 | 0.1845 | **0.1803** |
>
>
> **W6. Explanation of the New Paradigm and Performance**
>
> Our method **is the first** to approach the ASSR task from a signal reconstruction perspective, constructing a **continuous high-resolution (HR) signal in RGB space**. This unique paradigm enables us to generate a single Gaussian field and then **rapidly render images at any scale**, meeting various upscaling demands with just one generation process. In comparison, previous methods like GSASR cannot reconstruct a continuous RGB HR signal and must regenerate for each scale, resulting in lower efficiency and performance.
>
> **Speed is a major highlight of our approach.** As shown in Table 2, when rendering across multiple scales, GSASR requires an average of **89ms** per inference, whereas our method takes only **4.6ms**—a dramatic improvement that demonstrates our method's superior efficiency. Additionally, our approach achieves superior PSNR, SSIM, LPIPS, FID, and DISTS scores, further validating its effectiveness.

---

> ### Author Response · Authors · 2025-11-15
> **Reply3**
>
> **W7. More Ablation Studies on DDCW**
>
> We **have already conducted comprehensive ablation experiments on DDCW in the main paper (see Table 4)**, where the setting "without DDCW" means removing the dictionary and directly learning covariance matrices from data. In this configuration, the network fails to converge and cannot produce meaningful reconstructions. This result aligns with our analysis in Figure 2, which demonstrates that directly optimizing Gaussian parameters without DGP guidance leads to unstable training and local minima. Thus, DDCW is **not just beneficial—it's essential** for effective and stable optimization in Gaussian space.
>
> To facilitate your review, we have also included additional ablation results for DDCW in Table 5 of this rebuttal. The results further confirm the importance of DDCW: without it, PSNR drops dramatically, underscoring the critical role DDCW plays in our method.
>
> **Table 5. Ablation Study on DDCW**
>
> | DDCW | PSNR |
> | --- | --- |
> | w    | **28.2** |
> | w/o  | 12.3     |
>
> These findings reinforce that DDCW is indispensable for achieving high-quality reconstruction and stable training.
>
>
> **Q2. Further Explanation of DGP**
>
> The Deep Gaussian Prior (DGP) is a fixed prior that remains unchanged throughout training; it is **not fine-tuned** during our process. DGP is extracted once from large-scale natural image statistics and acts as a stable reference for modeling covariances. This fixed prior provides crucial regularization, guiding the network toward meaningful solutions.
>
> In our preliminary experiments, we attempted to learn covariance matrices directly without using the predefined DGP dictionary. However, this approach resulted in non-convergence and poor reconstruction quality. These results highlight the importance of DGP: by serving as a consistent and informative prior, it ensures stable optimization and effective learning in our framework.
>
>
> **Q4. Ablation Study on Gaussian Kernels**
>
> We appreciate your suggestion regarding the number of Gaussian kernels per pixel. **In fact, we have already conducted this experiment and reported the results in the supplementary materials (see Table 13, page 18, Appendix A.4)**. In our implementation, each low-resolution pixel is represented by four Gaussian kernels. We performed an ablation study to evaluate the impact of using different numbers of Gaussians per pixel (1, 4, and 9).
>
> As shown in Table 13 and Table 6 of this rebuttal, using four Gaussians achieves the best balance between reconstruction quality and computational efficiency. Increasing the number beyond four provides only marginal improvements in PSNR while introducing unnecessary computational complexity. These results confirm that four Gaussians per pixel offer the optimal trade-off in our framework.
>
> This observation is consistent with your suggestion and our own analysis. Additionally, we discuss this topic further in Appendix A.9 (Limitation and Future Work), noting that a fixed number of Gaussians may cause redundancy in low-frequency regions. As mentioned, a promising future direction is to adopt adaptive allocation strategies, assigning more Gaussians to regions with complex textures and fewer to smoother areas.
>
> **Table 6. Ablation Results for Number of Gaussian Kernels per Pixel**
>
> | Number of Gaussians per Pixel (N) | PSNR   | SSIM    |
> | ---------------------------------- | ------ | ------- |
> | 1                                  | 28.01  | 0.8254  |
> | 4                                  | **28.22** | **0.8292** |
> | 9                                  | 28.18  | 0.8284  |
>
>
> **Q7. Additional Details**
>
> The reported 1 ms rendering time was measured on an NVIDIA V100 GPU, using the same rasterization implementation as GaussianImage. This timing corresponds to generating ×4 outputs from 128 × 128 low-resolution inputs. For extremely high-resolution outputs (such as 4K or 8K), the rendering time increases roughly linearly with the number of target pixels; however, it remains significantly faster than conventional neural decoding pipelines.
>
> In our experiments, even 4K rendering completes in approximately 5 ms on a single V100 GPU, maintaining near-instantaneous performance. Importantly, memory usage scales with the number of Gaussians, not the output resolution, so we did not observe any practical memory or precision limitations during rendering.
>
>
> **Q8. Code and Model Release Commitment**
>
> Please rest assured that **we are fully committed to open-sourcing all code and model weights** associated with this work. Upon acceptance, we will release the complete implementation and pretrained models to the community, ensuring that researchers and practitioners can easily reproduce, validate, and build upon our results. We believe this will facilitate further progress and collaboration in the field.
>
>
> Thank you again for your valuable comments, which have greatly helped improve our paper. If you have any further suggestions, please feel free to let us know.

---

> > ### Comment · Reviewer_hqg6 · 2025-11-21
> >
> > The author response addresses the majority of the earlier technical concerns, including the relationship to GSASR/GaussianSR, the role and implementation of DGP and DDCW, the details of CGM and APD, the number of Gaussians per pixel, runtime behavior at high resolutions, and reproducibility. The additional BRATS evaluation also partially alleviates concerns about generalization beyond standard natural-image benchmarks.
> >
> > The remaining points of concern listed below are largely about framing rather than missing experiments:
> > - the “new paradigm” language still feels stronger than warranted by the relatively modest PSNR gains over GSASR and could be tempered to emphasize practical efficiency and engineering contributions;
> >
> > - the complexity and heavily engineered nature of the pipeline would benefit from explicit acknowledgment as a design trade-off; and
> >
> > - several important clarifications from the rebuttal (fixed DGP, CGM/MLP details, APD offset bounds, DDCW ablation, Gaussians-per-pixel ablation) should be integrated into the main text or appendix.
> >
> > Subject to these adjustments in positioning and presentation, the work appears technically sound and practically valuable.

---

> > > ### Author Response · Authors · 2025-11-22
> > > **Thank You for Acknowledging and Supporting Our Work**
> > >
> > > We sincerely thank the reviewer for the positive assessment and for acknowledging that **most of the previous technical concerns have been satisfactorily addressed**. We greatly appreciate the constructive suggestions on framing and presentation, and we have revised the manuscript accordingly.
> > >
> > >
> > > 1. Thank you for this helpful suggestion. Following your advice, we have carefully revised the wording in the main text to more clearly emphasize practical efficiency and engineering contributions. We appreciate your feedback, which has meaningfully improved the presentation of our paper.
> > >
> > >
> > > 2. We appreciate the reviewer’s insightful comment. In future work, we will further streamline the overall pipeline and improve the network efficiency to make the framework more lightweight and easier to deploy across a wider range of real-world applications. The related discussion has also been included in Section 6 of the main text.
> > >
> > >
> > > 3. We sincerely appreciate the reviewer’s valuable suggestions, which have greatly helped us improve the clarity and completeness of the work. Following these recommendations, all relevant experiments and analyses have been added to the main paper or the supplementary material in their appropriate locations. Specifically:
> > >
> > >
> > > - Fixed DGP: added to Section *4.2* (main text).
> > > - CGM / MLP implementation details: added to Section *A.5* (supplementary).
> > > - APD offset bounds and their motivation: added to Section *A.5* (supplementary).
> > > - DDCW ablation study: added to Section *5.3* (main text).
> > > - Gaussians-per-pixel ablation: added to Section *A.5* (supplementary).
> > >
> > > These revisions improve clarity and directly address the reviewer’s concerns regarding reproducibility and transparency, with all corresponding updates highlighted in blue in the revised manuscript.
> > >
> > > ---
> > >
> > > We sincerely appreciate the reviewer’s positive recognition that “**the work appears technically sound and practically valuable.**” We are glad that our revisions and responses have fully addressed your remaining concerns, and your thoughtful suggestions have significantly improved the clarity and precision of the manuscript. We genuinely hope that, in your final evaluation, you may consider giving the work a positive assessment, and we truly appreciate your encouraging acknowledgement of its technical soundness and practical value.

---

### Comment · Reviewer_mpzJ · 2025-11-20
**Reply by Reviewer mpzJ**

I think the paper makes solid contributions in continuous modeling, the DGP design, and the unified arbitrary-scale SR framework. After reading the authors’ responses for me and other reviewers. I’m keeping my original rating.

---

> ### Author Response · Authors · 2025-11-20
> **We sincerely appreciate your positive assessment of our work and are grateful for the time and thoughtful feedback you provided.**
>
> We sincerely appreciate your positive assessment of our work and are grateful for the time and thoughtful feedback you provided.

---

### Author Response · Authors · 2025-11-30
**Summary**

We sincerely thank the program chairs, senior area chairs, area chairs, and reviewers (**R1-hqg6, R2-mpzJ, R3-Jk3f, R4-wJ6J**) for their time, detailed feedback, and thoughtful discussion throughout the review process. To facilitate the final decision, we provide the following summary of the rebuttal discussion and the current status of reviewer feedback.

---

## Summary of Rebuttal-Discussion

Overall, reviewers acknowledged the contributions and practical value of our work. They highlighted its **novelty** (R1, R2, R3, R4), **fast arbitrary-scale rendering** (R2, R3, R4), **strong memory efficiency** (R1, R4), **modeling flexibility through Gaussian splatting** (R2, R3), and **elegant design of the one-pass continuous framework** (R2, R3).
 During the discussion phase, we addressed all raised concerns with detailed clarifications and new experiments. The main discussion points and our corresponding responses are summarized below:

1. **Clarifying the novelty and conceptual contribution of our Pixel-to-Gaussian paradigm**
   We emphasized that unlike GSASR or GaussianSR, our method reconstructs a continuous high-resolution RGB signal in a single pass, requiring no scale-specific decoding. This enables arbitrary-scale rendering with ~1ms latency and high memory efficiency. *(R1, R2, R3)*

2. **Explaining the role and construction of the Deep Gaussian Prior (DGP) and DDPW**
   DGP is a fixed statistical prior derived from 40,000 natural images, offering strong regularization for Gaussian optimization. Ablations showed that removing DDCW leads to training collapse, confirming its necessity. *(R1, R2, R3)*

3. **Ablation studies on APD and CGM modules**
   We conducted ablations on both Adaptive Position Drifting (APD) and Color Gaussian Mapping (CGM), demonstrating their effectiveness in enhancing spatial alignment and color reconstruction. *(R1)*

4. **Demonstrating generalization in real-world and OOD scenarios**
   New experiments on COZ (real-world ASSR) and BRATS (medical images) verified strong generalization and robustness without adjusting DGP. *(R1, R2, R3)*

5. **Clarifying training strategy and runtime reporting**
   We explained that LR sizes are fixed during training, enabling efficient parallelization without padding. We also clarified single-scale and multi-scale runtime measurements and provided GPU memory usage comparisons. *(R4)*


---

## Current Status of Reviewer Feedback

1. **R1 (Initial Score: 4)** acknowledged that all major technical concerns—including distinctions from prior work, stability of DGP/DDCW, ablations, and generalization tests—as well as concerns regarding wording and framing have been resolved. The reviewer explicitly stated that **our responses addressed earlier issues**, adding that **“the work appears technically sound and practically valuable.”**

2. **R2 (Initial Score: 8)** gave high recognition to our work, highlighting its **novel continuous modeling**, the **Deep Gaussian Prior design**, and the **overall speed and practicality** of the framework. After clarification on DGP construction, Gaussian kernel selection, and real-world generalization, the reviewer expressed **full satisfaction with no remaining concerns** and explicitly **recommended acceptance**.

3. **R3 (Initial Score: 6)** appreciated our **modeling strategy, efficiency, and extensibility**. Their theoretical questions were fully addressed during the rebuttal, and the reviewer confirmed that **all issues were resolved** and maintained a **positive recommendation**.

4. **R4 (Initial Score: 6)** recognized the **practical value and robustness** of our method. Concerns about training strategy, runtime, and memory usage were **all addressed**, and the reviewer **recommended acceptance**.






---

## Overall

Our paper introduces a unified and efficient solution to arbitrary-scale super-resolution by reformulating it as continuous RGB signal reconstruction with 2D Gaussian modeling. This design enables ultra-fast rendering, stable training, and strong generalization across synthetic and real-world domains. All reviewers who joined the discussion confirmed that **their concerns were resolved**. All reviewers have acknowledged the value of our work, resulting in an **average score of 6**. Specifically, three reviewers now provide scores of **6 or 8**, reflecting a converging positive assessment of the method’s technical soundness and practical value.

We hope this summary assists your final decision. We sincerely appreciate your time and thoughtful evaluation.

Best regards,
Authors

---

### Meta-Review · Area_Chair_LTyN · 2026-01-02

**Summary:**

This paper proposes an effective continuous super-resolution with 2D gaussian modeling. The major concerns include the limited novelty, heavily engineered solution, marginal quantitative gains, dependence on the prior dictionary, and unclear generalization ability.

**Reviewer Concerns:**

The major concerns including the limited novelty, heavily engineered solution, marginal quantitative gains, dependence on the prior dictionary, and unclear generalization ability should be addressed. In the rebuttal, the authors solve the most concerns of reviewers. However, the concerns w.r.t. heavily engineered solution and marginal quantitative gains are not solved well. In addition, the differences from the GaussianSR and GSASR should be better clarified.

**Reviewer Scores:**

As mentioned above, the authors solve the most concerns of reviewers. However, the concerns w.r.t. heavily engineered solution and marginal quantitative gains are not solved well. In addition, the differences from the GaussianSR and GSASR should be better clarified.

---

### Decision · Program_Chairs · 2026-01-26

Accept (Poster)